# Training Vision-Language Transformers from Captions

**Liangke Gui**[*1]      **Yingshan Chang**[*1]      **Qiuyuan Huang**[2]      **Subhojit Som**[2]
**Alex Hauptmann**[1]      **Jianfeng Gao**[2]      **Yonatan Bisk**[1]
[1]Carnegie Mellon University      [2]Microsoft Research
**github.com/guilk/VLC**

**Reviewed on OpenReview:** `https://openreview.net/forum?id=xLnbSpozWS`

## Abstract

Vision-Language Transformers can be learned without low-level human labels (e.g. class labels, bounding boxes, etc). Existing work, whether explicitly utilizing bounding boxes (Chen et al., 2020b; Tan & Bansal, 2019; Lu et al., 2019) or patches (Kim et al., 2021), assumes that the visual backbone must first be trained on ImageNet (Russakovsky et al., 2015) class prediction before being integrated into a multimodal linguistic pipeline. We show that this is not necessary and introduce a new model **V**ision-**L**anguage from **C**aptions (**VLC**) built on top of Masked Auto-Encoders (He et al., 2022) that does not require this supervision. We seek to provide general advice on multimodal pretraining by examining the roles of (a) unimodal initialization, (b) unimodal architectural components and (c) data annotation in the pretraining corpus. Our extensive and carefully controlled studies suggest that none of the above factors is absolutely important in achieving versatile vision-language representations. We conclude our analysis with suggestions on the choices of initialization, architectural components, and annotation formats targeting a better balance between data efficiency and representation quality.

## 1 Introduction

Should vision guide language understanding or does language structure visual representations? Vision-language (VL) transformers have put language first. Most popular vision-language transformers (Chen et al., 2020b; Tan & Bansal, 2019; Li et al., 2019; Lu et al., 2019) only integrate vision from selected bounding boxes extracted by pretrained ImageNet (Russakovsky et al., 2015) classifiers. In this paradigm, the bag of visual tokens are embedded into an existing linguistic space (*i.e.*, the lexical embeddings of BERT (Devlin et al., 2019)).

The introduction of ViT (Dosovitskiy et al., 2021) empowered the community to flip the paradigm. Notably, ViLT (Kim et al., 2021) initializes with ViT, so it places visual representations as the initial conceptual space to which language must adhere. Additionally, there are engineering benefits to this paradigm as it removes the computationally expensive need

*. Equal Contribution.

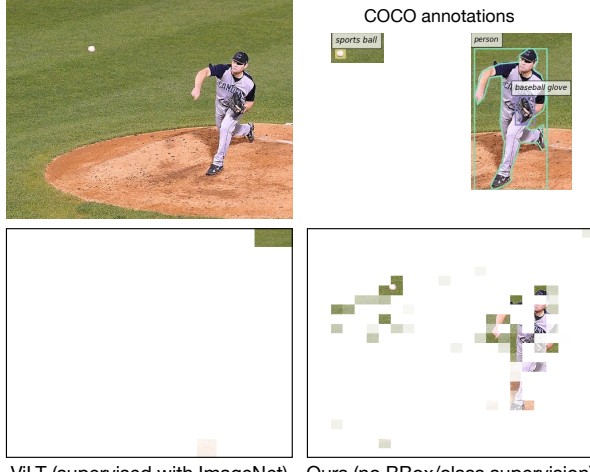

*A pitcher at a baseball game who has just **thrown** the ball.*

ViLT (supervised with ImageNet)    Ours (no BBox/class supervision)

Figure 1: Aligned patches with the word **thrown**. Our model (**VLC**) produces a more meaningful distribution over the patches. We argue for the following factors towards learning such faithfully-aligned and flexible vision-language representations: (1) Masked Image Modeling, (2) Avoiding a vision-only processing stack, (3) Avoiding priors from ImageNet pretraining.

for Region of Interest (ROI) extraction. However,
because ViT is trained with supervised ImageNet labels, its representation may be constrained by the finite label space. The ImageNet concept space is still somewhat linguistic in nature when initialized, and requires expensive annotation, a hindrance to scaling to arbitrarily many visual categories.

We take the important next step and remove the need for supervised pretraining. An unsupervised visual semantics is learned via Masked Auto-Encoders (He et al., 2022) before language is integrated. This leads to both a better performing and more general model. In addition, every component can be improved and scaled with unsupervised and weakly aligned data – removing the need for future annotation efforts while still scaling to open-vocabulary domains in the wild.

Our **V**ision-**L**anguage from **C**aptions (**VLC**) model matches or outperforms nearly all vision-language transformers despite being 1. Smaller and faster at inference, 2. Avoiding use of ROIs, and 3. Not leveraging object-level supervised labels for pretraining. For downstream usage, token representations can be naturally adapted to producing class-labels **and** bounding-boxes, with moderate finetuning. We demonstrate strong performance of **VLC** across a comprehensive set of benchmarks covering V+L understanding, retrieval, reasoning and grounding. Performance also continues to improve with data and model size scaling, and as it relies only on weak alignment of image-text pairs. Our ablation study shows that key to consistence improvement on downstream tasks is the masked image modeling objective, which is in sharp contrast to existing findings. Finally, we provide several analyses on the underlying representations to understand what our models are learning and guide future V+L transformer research.

## 2 Related Work

**Vision-Language Modeling.** Based on how they encode images, most existing work on vision-language modeling fall into three categories. The first category (Lu et al., 2019; Tan & Bansal, 2019; Li et al., 2019; Su et al., 2020; Chen et al., 2020b; Li et al., 2020; Zhang et al., 2021; Li et al., 2021b; Qi et al., 2020) focuses on using pre-trained object detectors to extract region-level visual features (e.g., by Faster R-CNN (Ren et al., 2015)). Extracting region-level features requires pretrained object detectors with high-resolution inputs that can be time-consuming. To tackle these two issues, the second category (Huang et al., 2020; 2021; Jia et al., 2021) proposes to encode images by using grid features from convolutional neural networks. The third category (Dou et al., 2022a; Xue et al., 2021; Li et al., 2021a; Radford et al., 2021) uses a Vision Transformer (ViT) (Dosovitskiy et al., 2021) as the image encoder and designs different objective functions for vision-language pretraining. Such approaches employ an encoder-decoder architecture coupled with image captioning as their pretraining tasks. Typically, PaLI (Chen et al., 2023) and BLIP (Li et al., 2022; 2023) adopt the pretrained ViT as their visual components. Flamingo (Alayrac et al., 2022) initializes its visual encoder from NFNet (Brock et al., 2021). These models not only require substantial computation already consumed by their initialization checkpoints, but have also oriented their focus towards the ultimate performance by relying on even larger multimodal training datasets. CoCA (Yu et al., 2022) and SimVLM (Wang et al., 2022c) are similar to our approach in that they do not initialize from well-trained unimodal components. But both works adopt the encoder-decoder architecture where the decoder only outputs text, which we believe cannot naturally handle grounding tasks. CLIPPO (Tschannen et al., 2023) unifies image and text representation by rendering text as images. While such a representation scheme unlocks the possibility of scene-text reading, its performances on GLUE, VQA and image-text retrieval still lag far behind models that encode text from tokens. The most similar work to ours is ViLT (Kim et al., 2021). ViLT does not use pretrained object detectors or extra visual embedders for visual embedding, but still needs weights pretrained on ImageNet-21K for initialization. Another related study, VirTex (Desai & Johnson, 2021), aligns with a similar objective as ours, which is to train vision-language representations with reduced data annotation requirements. Differing from prior research, we offer a re-examination of the roles played by (a) unimodal initialization, (b) unimodal architectural components and (c) annotated pretraining data. With careful controls on model and data sizes, our results suggest that none of the above factors is absolutely important in achieving versatile vision-language representations. Additionally, our model exhibits superior performance when naturally extending to reasoning and grounding tasks that necessitate distinguishing part-whole relationships.

**Masked Language Modeling.** Masked language modeling (MLM) and its auto-regressive counterparts are widely used in natural language processing for learning text representations. MLM (Devlin et al., 2019; Tay et al., 2022) trains a model to predict a random sample of input tokens that have been masked in a multi-class setting. In vision-language pretraining, MLM has shown useful to enforce the consistency across modalities (Dou et al., 2022a; Zhang et al., 2021; Li et al., 2020; Kim et al., 2021). In vision-language modeling, we randomly mask some of the input tokens, and the model is trained to reconstruct the original tokens given the masked tokens and their corresponding visual inputs. To be consistent with previous work, we follow the default settings of training BERT (Devlin et al., 2019) for masked language modeling.

**Masked Image Modeling.** Masked image modeling (MIM) is a pretext task to learn representations from images corrupted by masking. Inspired by the success of masked language modeling (MLM) in NLP, different masked prediction objectives have been proposed for image tasks. iGPT (Chen et al., 2020a) predicts unknown pixels of a sequence. ViT (Dosovitskiy et al., 2021) predicts mean colors of masked patches. BEiT (Bao et al., 2022) proposes to use a pre-trained discrete variational autoencoder (dVAE) (Ramesh et al., 2021) to encode masked patches. MaskFeat (Wei et al., 2021) predicts HoG (Dalal & Triggs, 2005) features of the masked image regions. SimMIM (Xie et al., 2022) and MAE (He et al., 2022) predict RGB values of raw pixels by direct regression. MIM has also been explored in the field of vision-language representation learning by either regressing the masked feature values (Tan & Bansal, 2019; Chen et al., 2020b; Dou et al., 2022a; Kim et al., 2021) or predicting a distribution over semantic classes for the corresponding image region (Chen et al., 2020b; Lu et al., 2019; Su et al., 2020). In contrast to previous approaches (Chen et al., 2020b; Kim et al., 2021; Dou et al., 2022a) that show MIM does not contribute to or hurt the performance on downstream tasks, we show that using MIM can consistently improve the performance as the training steps increase.

## 2.1 Model Architecture

Our aim is a vision-language transformer that can be trained without the need for expensive object-level supervised labels (*e.g.*, class labels or object bounding boxes). More concretely, our results empirically show that such object based supervised signals are *not* necessary for vision-language pretraining. To this end, we use a ViT-based framework to learn multi-modal representations by 1) intra-modal reconstruction through masked image/language modeling; 2) inter-modal alignment through image-text matching. The architecture of our proposed **VLC** framework is illustrated in Figure 2. **VLC** consists of a modality-specific projection module 3.1, a multi-modal encoder 3.2 and three task-specific decoders 3.3. We aim for minimal visual and textual embedding designs during pretraining.

# 3 Method

## 3.1 Modality-specific Projection Module

While most of existing methods rely on complex ResNeXt (Huang et al., 2020) or object detection components (Chen et al., 2020b; Lu et al., 2019; Zhang et al., 2021; Li et al., 2020), we use a trainable *linear projection* layer to map flattened visual patches to the visual embedding space. The patch embeddings are represented as $\mathbf{v} = \{v_1, ..., v_K\} \in \mathbb{R}^{K \times d}$, where $K$ is the number of image patches and $d$ is the hidden dimension of our model. For text embedder, we follow BERT (Devlin et al., 2019) to tokenize the input sentence into WordPieces (Wu et al., 2016). We then adopt a word embedding lookup layer to project tokenized words to the textual embedding space. Here we use $\mathbf{w} = \{w_{CLS}, w_1, ..., w_T\} \in \mathbb{R}^{T \times d}$ to represent the token embeddings, where $T$ is the number of tokens and the special token `CLS` denotes the start of the token sequence. We encode patch and token positions separately by $v^{pos} \in \mathbb{R}^{1 \times d}$ and $w^{pos} \in \mathbb{R}^{1 \times d}$. We use $v^{type} \in \mathbb{R}^{1 \times d}$ and $w^{type} \in \mathbb{R}^{1 \times d}$ as modality-type embeddings to distinguish the modality difference between patch and token embeddings. The final representations of each patch $v_i$ and token $w_j$ are calculated as

$$\hat{v}_i = \text{LayerNorm}(v_i + v_i^{pos} + v^{type}), \quad \text{and} \tag{1}$$

$$\hat{w}_j = \text{LayerNorm}(w_j + w_j^{pos} + w^{type}). \tag{2}$$

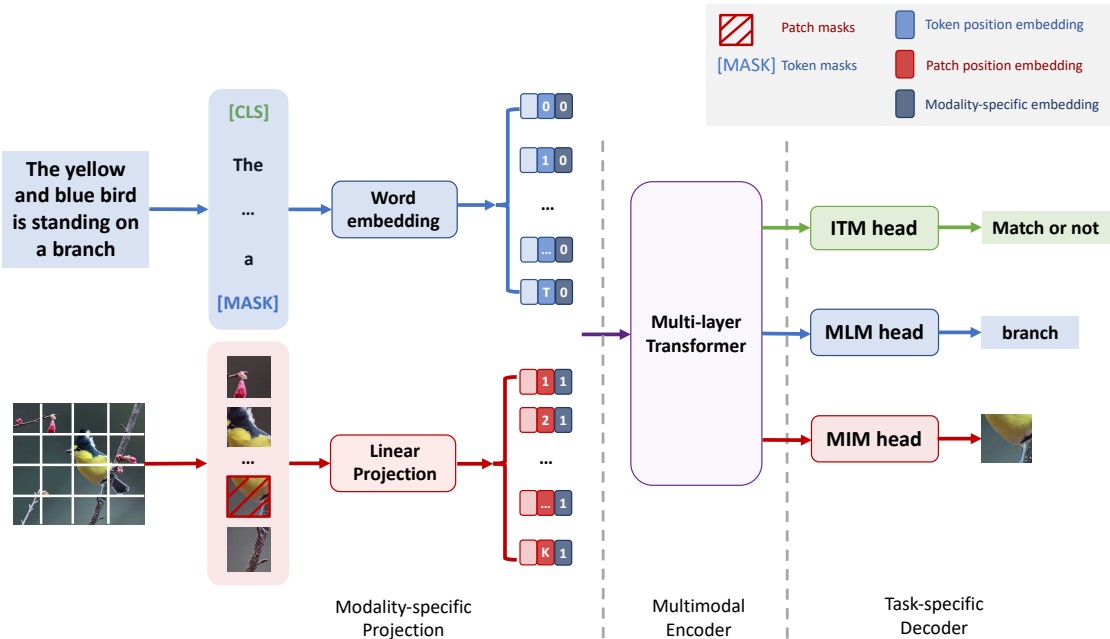

Figure 2: The overall architecture of **VLC**. Our model consists of three modules: (1) Modality-specific projection. A simple linear layer projects image patches and token embeddings into the same representation space; (2) Multi-modal encoder. We use a 12-layer ViT (Dosovitskiy et al., 2021) initialized from MAE (He et al., 2022) (ImageNet-1K without labels) as our Transformer backbone; (3) Task-specific decoder. We pretrain multi-modal representations with masked image/language modeling (MIM/MLM) and image-text matching objectives. This work justifies the importance of the MIM objective throughout V+L pretraining, in addition to providing good initializations for the Transformer encoder.

## 3.2 Multi-modal Encoder

To learn the contextual representations from both visual and textual modality, we follow single-stream approaches (Kim et al., 2021; Chen et al., 2020b) and use the ViT-B/16 architecture as our multi-modal encoder. ViT-B/16 consists 12 alternating layers of multiheaded self-attention (MSA) and MLP blocks. LayerNorm comes before every block and residual connections after every block (Dosovitskiy et al., 2021). We use a merged-attention (Dou et al., 2022a) mechanism to fuse the visual and textual modalities. More specifically, we concatenate the token and patch embeddings together as $\{\hat{w}_{CLS}, \hat{w}_1, ..., \hat{w}_T, \hat{v}_1, ..., \hat{v}_K\}$, then feed them into the transformer. We use the hidden states $h$ at the output of the last layer of the encoder as the contextual representations $\{h_{CLS}, h_1^w, ..., h_T^w, h_1^v, ..., h_K^v\}$. In sharp contrast to existing approaches that use object detectors, visual detectors pretrained with supervised labels or pretrained language models (*e.g.*, BERT, Roberta), we initialize our model with MAE pretrained on ImageNet-1K with no labels.

## 3.3 Pretraining Objectives

To learn a universal visual and textual representation for vision-and-language tasks, we apply self-supervised methods to pre-train a model on a large aggregated dataset. Unlike previous approaches that only mask text tokens, we randomly mask both image patches and text tokens simultaneously. We train our model with masked image modeling (MIM), masked language modeling (MLM), and image-text matching (ITM).

**Masked Language Modeling.** In language pretraining, MLM randomly masks input tokens, and the model is trained to reconstruct the original tokens based on unmasked context. Following BERT (Devlin et al., 2019), we randomly mask text tokens with a probability of 0.15, and replace the masked ones $\mathbf{w_m}$ with a special token `[MASK]`. The goal is to predict the masked tokens based on both non-masked text tokens $\mathbf{w_{\setminus m}}$

and image patches $\mathbf{v}_{\backslash\mathbf{m}}$. The learning target $\mathcal{L}_{MLM}$ can be formulated as

$$\mathcal{L}_{MLM} = -\mathbb{E}_{(\mathbf{w},\mathbf{v})\sim D}\log p(\mathbf{w_m}|\mathbf{w}_{\backslash\mathbf{m}},\mathbf{v}_{\backslash\mathbf{m}}). \tag{3}$$

We use a linear layer with default parameters (Devlin et al., 2019) as the MLM head to output logits over the vocabulary, which are used to compute the negative log likelihood loss for the masked text tokens.

**Masked Image Modeling.** Existing approaches explore MIM either by regressing the masked features values (Chen et al., 2020b; Kim et al., 2021; Xue et al., 2021) or by predicting a distribution over semantic classes for an image region (Chen et al., 2020b; Lu et al., 2019; Dou et al., 2022a). In contrast, we randomly mask image patches with a probability of 0.6, and reconstruct the missing pixels based on both non-masked tokens $\mathbf{w}_{\backslash\mathbf{m}}$ and patches $\mathbf{v}_{\backslash\mathbf{m}}$ (as in MAE (He et al., 2022)). The learning target $\mathcal{L}_{MIM}$ can be formulated as

$$\mathcal{L}_{MIM} = \mathbb{E}_{(\mathbf{w},\mathbf{v})\sim D}f(\mathbf{v_m}|\mathbf{w}_{\backslash\mathbf{m}},\mathbf{v}_{\backslash\mathbf{m}}), \tag{4}$$

where the feature regression objective $f$ is to regress the masked image patch representations to pixel values. We use an 8-layer transformer as the MIM head $r$. For a masked image patch $v_i$, the objective $f$ can be formulated as: $f(v_i|\mathbf{w}_{\backslash\mathbf{m}},\mathbf{v}_{\backslash\mathbf{m}}) = ||r(h_i^v) - v_i||^2$. Each output of the MIM head is a vector of pixels representing a patch. Different from the observations in ViLT (Kim et al., 2021) and METER (Dou et al., 2022a), we find that MIM consistently improves the performance on downstream tasks as training progresses.

**Image-Text Matching.** Given a batch of image and text pairs, the ITM head identifies if the sampled pair is aligned. We randomly replace the aligned image with a different one with a probability of 0.5. We use the special token [CLS] as the fused representation of both modalities, and feed $h_{CLS}$ to the ITM head. The learning target $\mathcal{L}_{ITM}$ can be formulated as

$$\mathcal{L}_{ITM} = -\mathbb{E}_{(\mathbf{w},\mathbf{v})\sim D}\log p(y|\mathbf{w},\mathbf{v}), \tag{5}$$

Where $y \in \{0,1\}$ indicates whether the image and text are matched ($y = 1$) or not ($y = 0$). We use a single linear layer as the ITM head and compute negative log likelihood loss as our ITM loss. We weight the pretraining objectives equally so the full pre-training objective is:

$$\mathcal{L} = \mathcal{L}_{MLM} + \mathcal{L}_{ITM} + \mathcal{L}_{MIM} \tag{6}$$

For a fair comparison with existing approaches, We do not include image-text contrastive loss (Li et al., 2021a; 2022), momentum distillation (Li et al., 2021a) and other techniques in our implementation.

## 4 Experimental Setup

### 4.1 Pre-training Datasets

Following previous work (Chen et al., 2020b; Kim et al., 2021; Li et al., 2021a; Dou et al., 2022a), our pre-training corpus comprises four commonly used vision-language datasets including COCO (Lin et al., 2014), Visual Genome (Krishna et al., 2017), Google Conceptual Captions (Sharma et al., 2018) and SBU Captions (Ordonez et al., 2011), totalling 4.0M unique images and 5.1M image-text pairs. To show the benefits of data-scaling, we also use the VinVL (Zhang et al., 2021) pretraining data which includes Flickr30k (Young et al., 2014), GQA (Hudson & Manning, 2019), VQA (Goyal et al., 2017), VG-QAs (Krishna et al., 2017) and a subset of OpenImages (Krasin et al., 2016). This larger pre-training corpus contains 5.65M unique images (see detailed statistics in Appendix A). We carefully control the amount of pretraining data used for both the initialization checkpoint (*e.g.*, ImageNet (Russakovsky et al., 2015), MSCOCO Lin et al. (2014), WebLI Chen et al. (2023)) and for vision-language pretraining. Note that many recent works tend to underestimate the extent of data and computational resources necessary for their unimodal initializations. Consequently, this oversight inadvertently obscures the fact that their final outcomes are heavily reliant on considerably more resources than would constitute a fair comparison. Additional analysis can be found in Appendix B.

### 4.1.1 Fine-tuning Datasets

We fine-tune our model on **image-text retrieval:** Flickr30K (Plummer et al., 2015) and MSCOCO (Lin et al., 2014), **image-text understanding:** VQAv2 (Goyal et al., 2017) and NLVR$^2$ (Suhr et al., 2018), and **image-text grounding:** Refcoco (Yu et al., 2016; Mao et al., 2016) tasks. We fine-tune all encoder layers together with task-specific heads. For retrieval tasks, we follow the standard splits and evaluate our models in the finetuning settings. For VQAv2, we follow the standard practice (Chen et al., 2020b; Li et al., 2021a) to train the models with both training, validation and additional question-answer pairs from Visual Genome while reserving $1,000$ validation samples for internal validation. For Refcoco, we propose a new algorithm to obtain bounding box outputs based on affinities between patch and text encodings. This only requires fine-tuning the encoder backbone without introducing extra parameters. Finally, ablation studies validate our key design components. More experimental details can be found in Appendix C and D.

### 4.2 Implementation Details

We pretrain two variants of the multi-modal encoder which uses a 86M parameter ViT-B/16 denoted as **VLC**$_{\text{Base}}$ and 307M parameter ViT-L/16 denoted as **VLC**$_{\text{Large}}$. Both variants are initialized with MAE pre-trained on ImageNet-1K without labels. For text inputs, we tokenize text with the *bert-base-uncased* and *bert-large-uncased* tokenizer, respectively. The text embedding parameters are learned from scratch, in lieu of loading pre-trained BERT weights. We randomly mask image patches with a probability of 0.6 and text tokens with a probability 0.15. To accelerate training, we follow MAE (He et al., 2022) and skip the mask token [MASK] in the encoder and only apply it in the lightweight decoder. We use AdamW (Loshchilov & Hutter, 2018) with a weight decay of 0.01. The learning rate is warmed-up to $1e^{-4}$ in the first 10% of total training steps and is decayed to zero for the rest of the training following a linear schedule. During pre-training, we resize the shorter edge of input images to 384, take random image crops of resolution $384 \times 384$, and apply RandAugment (Cubuk et al., 2020). We pre-train for $200k$ steps with a batch size of $4,096$. For the parameter estimation, we exclude the textual embedder as it is shared by all vision-language transformers. We also exclude the parameters of all the auxiliary heads as they are only required during pretraining. Unless otherwise specified, we use the *base* version of **VLC** for downstream evaluation and visualization.

## 5 Downstream Experiments & Results

We evaluate the effectiveness of our pre-training strategy under both zero-shot and fine-tuning settings. **VLC** is competitive across a diverse set of standard V+L benchmarks, broadly categorized into Image-Text Retrieval, Image-Text Understanding, and Image-Text Grounding. In addition fine-tuning based evaluations, we also perform inference-only experiments on multiple challenging benchmarks that test reasoning (Thrush et al., 2022; Ji et al., 2022) and open-domain question answering (Schwenk et al., 2022).

### 5.1 Zero-shot Experiments

First we verify the strength of our approach in a zero-shot setting before investing in larger fine-tuning results. We adapt our model to perform zero-shot visual reasoning on the Kilogram (Ji et al., 2022) dataset. Kilogram challenges a model to recognize an abstract Tangram shape from a language description. Tangram shapes drastically differ from natural scenes as they only contain abstract and implicit visual clues. Hence, recognition demands more sophisticated reasoning about the interplay between visual and linguistic cues. The task is formulated into a 10-way classification problem where the model chooses the most relevant image given an abstract shape description. Specifically, we rank image relevance by the logits calculated by the pretrained image-text matching head. Table 1 shows that our model consistently outperforms ViLT across all input conditions. This sup-

| Input Condition | ViLT | Ours | Human |
|---|---|---|---|
| WHOLE+BLACK | 12.9 | **13.8** | 47.7 |
| PARTS+BLACK | 12.5 | **15.2** | 49.1 |
| WHOLE+COLOR | 11.7 | **13.9** | 49.5 |
| PARTS+COLOR | 10.7 | **13.5** | 63.0 |

Table 1: Zero-shot accuracy on Kilogram (dev), a challenging image-text matching task for abstract visual recognition. **VLC** consistently outperforms ViLT across all input conditions.

ports our design consideration that freeing up the model from priors of limited ImageNet concepts helps it

| Model | | Text Retrieval | | | | | | Image Retrieval | | | | | |
| --- | --- | --- | --- | --- | --- | --- | --- | --- | --- | --- | --- | --- | --- |
| | | Flickr30K (1K) | | | MSCOCO (5K) | | | Flickr30K (1K) | | | MSCOCO (5K) | | |
| | Params | @1 | @5 | @10 | @1 | @5 | @10 | @1 | @5 | @10 | @1 | @5 | @10 |
| ALBEF† (Li et al., 2021a) | 163M | 94.3 | 99.4 | 99.8 | 73.1 | 91.4 | 96.0 | **82.8** | **96.7** | **98.4** | 56.8 | 81.5 | 89.2 |
| VinVL$_{Large}$ (Zhang et al., 2021) | 452M | - | - | - | 75.4 | 92.9 | 96.2 | - | - | - | **58.8** | 83.5 | 90.3 |
| UNITER$_{Large}$ (Chen et al., 2020b) | 371M | 87.3 | 98.0 | 99.2 | 65.7 | 88.6 | 93.8 | 75.6 | 94.1 | 96.8 | 52.9 | 79.9 | 88.0 |
| METER-Swin$_{Base}$ (Li et al., 2020) | 288M | 92.4 | 99.0 | 99.5 | 76.2 | 93.2 | 96.8 | 79.0 | 95.6 | 98.0 | 54.9 | 81.4 | 89.3 |
| PixelBERT (Huang et al., 2020) | 144M | 87.0 | 98.9 | 99.5 | 63.6 | 87.5 | 93.6 | 71.5 | 92.1 | 95.8 | 50.1 | 77.6 | 86.2 |
| ViLT (Kim et al., 2021) | 86M | 83.5 | 96.7 | 98.6 | 61.5 | 86.3 | 92.7 | 64.4 | 88.7 | 93.8 | 42.7 | 72.9 | 83.1 |
| **VLC**$_{Base}$ (with 5.6M examples) | 86M | 89.2 | 99.2 | 99.8 | 71.3 | 91.2 | 95.8 | 72.4 | 93.4 | 96.5 | 50.7 | 78.9 | 88.0 |
| **VLC**$_{Large}$ (with 5.6M examples) | 307M | **94.4** | **99.6** | **99.9** | **76.7** | **94.5** | **97.3** | 79.1 | 95.8 | 98.2 | 58.4 | **84.0** | **91.1** |

Table 2: On Image-text retrieval tasks, we see substantial gains over state-of-the-art methods with bounding-box pretraining. †ALBEF uses specifically designed coarse-to-fine objectives for the image-text retrieval task.

generalize to arbitrarily many linguistically-specified visual categories. Further improvement may come from additional finetuning to familarize our model with the Tangram shape domain, which is left for future work.

## 5.2 Finetuning Experiments

Next we investigate the role that fine-tuning has on our downstream performance across traditional retrieval and understanding benchmarks, before analyzing training and performance curves.

### 5.2.1 Image-Text Retrieval

We begin with a proof of concept experiment, evaluating our model on the Karpathy splits of the Flickr30K (Plummer et al., 2015) and MSCOCO (Lin et al., 2014) benchmarks. Table 2 compares **VLC** to strong multimodal transformers which leverage ROIs, more parameters, and are pretrained on ImageNet classification. Note that as most of detection-based models have the advantage of using Faster R-CNN (Ren et al., 2015) pre-trained on VG (Krishna et al., 2017) or MSCOCO (Lin et al., 2014). The closest comparison to **VLC**$_{Base}$ is ViLT (same model size), though ViLT still requires more supervised data in the form of ImageNet pretraining for ViT (Dosovitskiy et al., 2021). When comparing to dual-encoder models, our **VLC**$_{Large}$ achieves competitive results across all settings. ALBEF uses pre-trained ViT and BERT model for initialization. Additionally, it specifically designs the coarse-to-fine objectives while we directly fine-tune the pre-trained ITM head for retrieval tasks. Thus we treat ALBEF as a strongest available baseline.

### 5.2.2 Image-Text Understanding

Table 3 presents results on two popular image-text understanding datasets: VQAv2 and NLVR$^2$. We use the same training data as ViLT denoted as 4M and VinVL denoted as 5.6M.

**Comparison to models supervised/initialized with ImageNet bounded boxes.** Most of these models use object detectors pretrained on VG (Krishna et al., 2017) or MSCOCO (Lin et al., 2014) to extract region features. Object detectors help in VQA tasks as they mainly ask about objects. Within the similar scale of pretraining data, our model achieves competitive performance on both tasks. Note that our model uses 384×384 or 576×576 as input resolution during our fine-tuning stages. This resolution is much lower compared with previous work using 800×1333 (Lu et al., 2019; Chen et al., 2020b). In particular, *VinVL* (Zhang et al., 2021) has a multi-stage pre-training for its object detector that has access to ImageNet-5K (Xie et al., 2017) (6.8M images from 5K classes) and four object detection datasets (Shao et al., 2019; Krishna et al., 2017; Lin et al., 2014; Krasin et al., 2016) (2.5M images with bounding box annotations). Our **VLC**$_{Large}$, which has a similar model size, achieves better performance on the VQA task and competitive results on the NLVR task without any supervised initialization.

---

ViLT uses ViT-B/32 pretrained with ImageNet-21K and finetuned on ImageNet-1K with supervised labels.

| Model | Param | VQAv2 test | | NLVR$^2$ | |
|---|---|---|---|---|---|
| | | -dev | -std | dev | test |
| *Supervised ImageNet Bounded Boxes* | | | | | |
| ViLBERT (Lu et al., 2019) | 274M | 70.55 | 70.92 | - | - |
| LXMERT (Tan & Bansal, 2019) | 240M | 72.42 | 72.54 | 74.90 | 74.50 |
| VisualBERT (Li et al., 2019) | 170M | 70.80 | 71.00 | 67.4 | 67.0 |
| UNITER$_{Large}$ (Chen et al., 2020b) | 371M | 73.82 | 74.02 | 79.12 | 79.98 |
| OSCAR$_{Large}$ (Li et al., 2020) | 371M | 73.61 | 73.82 | 79.12 | 80.37 |
| VinVL$_{Large}$ $^\dagger$ (Zhang et al., 2021) (5.6M) | 452M | 76.52 | 76.60 | **82.67** | **83.98** |
| *Supervised ImageNet Classes* | | | | | |
| METER-Swin$_{Base}$ $^\ddagger$ (Dou et al., 2022b) | 288M | 76.43 | 76.42 | 82.23 | 82.47 |
| ALBEF (Li et al., 2021a) | 163M | 74.54 | 74.70 | 80.24 | 80.50 |
| Visual Parsing (Xue et al., 2021) | 180M | 74.00 | 74.17 | 77.61 | 78.05 |
| PixelBERT (Huang et al., 2020) | 144M | 74.45 | 74.55 | 76.5 | 77.2 |
| ViLT (Kim et al., 2021) | 86M | 71.26 | - | 75.70 | 76.13 |
| *No supervised classes or bounding boxes* | | | | | |
| **VLC**$_{Base}$ (ours − 4M) | 86M | 72.98 | 73.03 | 77.04 | 78.51 |
| **VLC**$_{Base}$ (ours − 5.6M) | 86M | 74.02 | 74.0 | 77.70 | 79.04 |
| **VLC**$_{Large}$ (ours − 5.6M) | 307M | **76.95** | **77.02** | 82.27 | 83.52 |
| *Pre-trained or initialized with > 10M data* | | | | | |
| X-VLM (Zeng et al., 2022) (16M) | 216M | 78.22 | 78.37 | 84.41 | 84.76 |
| BLIP (Li et al., 2022) (129M) | 252M | 78.25 | 78.32 | 82.15 | 82.24 |
| OFA (Wang et al., 2022b) (54M) | 930M | 82.0 | 82.0 | - | - |
| CoCa (Yu et al., 2022) (4.8B) | 2.1B | 82.3 | 82.3 | 86.1 | 87.0 |

Table 3: Despite that **VLC** is only pretrained with weakly-aligned image-caption pairs, it matches or outperforms larger and more heavily supervised approaches within a similar scale of training data and model size. $^\dagger$VinVL uses the object detector trained with $6.8M$ labeled imageNet images and 2.5M images with bounding box annotations. $^\ddagger$METER-Swin$_{Base}$ uses Swin-B trained with $14M$ labeled ImageNet as the image encoder and pretrained Roberta as the text encoder.

**Comparison to models with supervised ImageNet classes.** Most of these approaches use additional visual embedders together with a pretrained BERT as their backbones. For example, ALBEF (Li et al., 2021a), Visual Parsing (Xue et al., 2021), PixelBERT (Huang et al., 2020) use pre-trained ViT-B/16, Swin transformer, ResNeXt-152 as their visual embedder, respectively. All these embedders are trained with labeled ImageNet data. In particular, *METER-Swin*$_{Base}$ uses Swin-B/16 pretrained with more than 14M labeled ImageNet22K images as the image encoder and pretrained Roberta as the text encoder. Experiments show that our model achieves better results than larger and more heavily supervised approaches.

Note that some models have access to a much larger scale of data during pretraining. For example, METER-CLIP-ViT$_{Base}$ uses CLIP as the image encoder which is trained with 400M image-text pairs. X-VLM (Zeng et al., 2022) uses the same image encoder as METER-Swin$_{Base}$ but with extra bounding box annotations and two times the training data. These models achieve higher performance, but comparisons are out of scope.

### 5.2.3 Image-Text Grounding

Since the model was not pretrained to generate bounding boxes, we propose a novel algorithm which produces a bounding box based on patch-text affinities in a single forward pass. This results in much less inference time and higher parameter efficiency compared with competitive pipelines tailored to bounding box generation. See Figure 3 (Left) for an overview of the proposed grounding workflow and Appendix G for details.

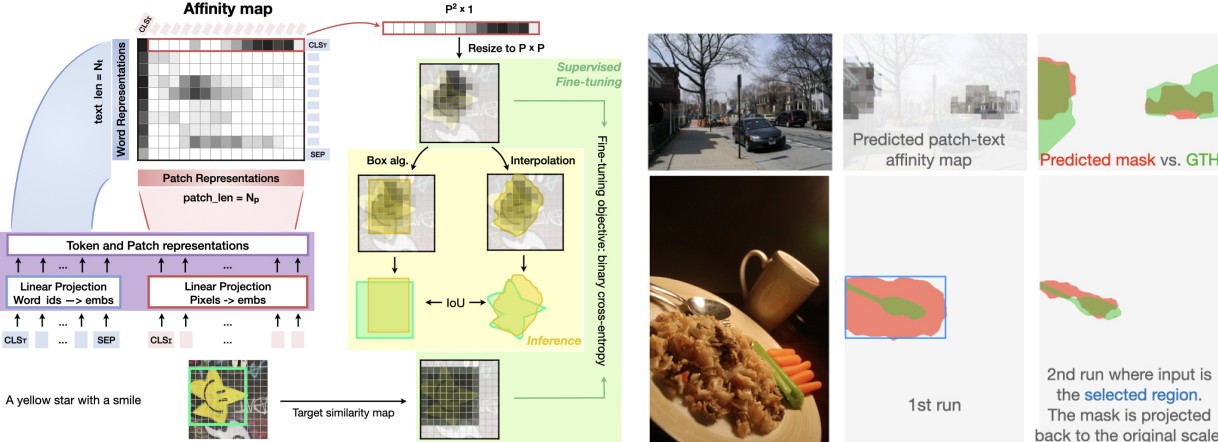

Figure 3: **Left:** Extending VLC to efficiently performing image-text grounding tasks: We finetune the Transformer backbone with supervision on similarities between patch and text representations. During inference, a bbox/mask can be obtained from the affinity scores with minimal computation overheads. **Right:** Reasonable segmentation masks resulted from interpolating patch-text affinities. Running the same inference procedure twice with coarse-to-fine resolution improves precision.

We finetune the encoder on the training sets of Refcoco, Refcoco+, and Refcocog (Yu et al., 2016; Mao et al., 2016) following the umd split. We convert ground-truth bounding boxes to patch-text affinity scores and use them to finetune the preceding representations. At inference time, we algorithmically predict a bounding box for each referential expression from affinity scores between the last layer patch and text representations.

**Comparison to modularized models** Previous models either use an RoI (Region-of-Interest) extractor to produce candidate bboxes to choose from (Chen et al., 2020b; Gan et al., 2020; Cho et al., 2021; Subramanian et al., 2022), or directly predict the coordinates and dimensions of bboxes in the image coordinate system (Deng et al., 2021). The former approach relies on an off-the-shelf detector, while the latter requires gating mechanisms to infuse linguistic information into a visual backbone. Our approach outperforms all previous approaches with modular designs (Table 4, Top), meanwhile achieving the best inference-time efficiency. This justifies our unified approach in which both understanding and localization tasks can benefit from large-scale representation learning.

**Comparison to unified models that have far more parameters or computation** Most of performant models under comparison incorporated bounding box annotations from VG (Krishna et al., 2017) in their pretraining data (Table 4, Bottom). With a much lighter architecture and much less annotated data, our model already achieves competitive performance. This holds promise that the performance will continue to improve as resolution and data are scaled up during pretraining.

**Generalization to dense prediction** Moreover, demonstrates that interpolating well-aligned path-text affinities naturally results in segmentation masks. Table 6 reports pixelwise-IoU between our interpolated masks and ground-truth masks on Refcoco/+/g. We have doubled the performance of TSEG (Strudel et al., 2022), the previous state-of-the-art method on Referring Expression Segmentation *without* training on ground-truth masks. Note, performing a second forward pass with zoomed-in regions leads to significantly finer mask contours (Figure 3 Right). We leave a closer investigation of this inference trick to future work.

| Model | Refcoco/+/g(umd) val | | |
|---|---|---|---|
| TSEG-CRF (Strudel et al., 2022) | 25.95 | 22.62 | 23.41 |
| **VLC**$_{\text{Base}}$ (ours) | 50.65 | 45.88 | 46.37 |

Table 6: We perform Referring Expression Segmentation through interpolating low-resolution affinity maps into segmentation masks without dense supervision.

| | Model | Time | Param | Refcoco/+/g(umd) val | | |
|---|---|---|---|---|---|---|
| Modular | VL-T5 (Cho et al., 2021) | 5.9x | 290M | — | — | 71.2 |
| | TransVG (Deng et al., 2021) | 2.8x | 169M | 80.83 | 68.00 | 68.71 |
| | TransVG++ (Deng et al., 2022) | — | — | 86.28 | 75.39 | 76.18 |
| | QRNet (Ye et al., 2022) | 5.5x | 273M | 84.01 | 72.94 | 73.03 |
| | RefTrans (Li & Sigal, 2021) | — | — | 85.65 | 77.55 | 79.25 |
| | SeqTR (Zhu et al., 2022) | 4.4x | 108M | 83.72 | 71.45 | 74.86 |
| | UNITER$_{\text{Large}}$ (Chen et al., 2020b) | — | 371M | 81.41 | 75.90 | 74.86 |
| | VILLA$_{\text{Large}}$ (Gan et al., 2020) | — | 371M | 82.39 | 76.17 | 76.18 |
| | **VLC**$_{\text{Base}}$ (ours) | 1.0x | 110M | 86.67 | 77.44 | 80.43 |
| Unified | OFA$_{\text{Medium}}$ (Wang et al., 2022a) | — | 93M | 85.34 | 76.09 | 78.76 |
| | Mdetr-R101 (Kamath et al., 2021) | 3.0x | 185M | 86.75 | 79.52 | 81.64 |
| | Mdetr-ENB3 (Kamath et al., 2021) | 2.6x | 153M | 87.51 | 81.13 | 83.35 |
| | UniTAB (Yang et al., 2022) | 19.4x | 198M | 88.59 | 80.97 | 84.58 |
| | OFA$_{\text{Base}}$ (Wang et al., 2022a) | 6.0x | 180M | 88.48 | 81.39 | 82.29 |
| | OFA$_{\text{Large}}$ (Wang et al., 2022a) | 11.5x | 470M | 90.05 | 85.80 | 85.89 |
| | OFA (Wang et al., 2022a) | — | 930M | 92.04 | 87.86 | 88.07 |

Table 4: Referring Expression Comprehension organized by: efficiency-performance and -versatility. *Modularized* baselines include bespoke components in their architecture. *Unified* baselines emphasize versatility over efficiency, and are pretrained with bounding box annotations. Our model is much lighter and faster. Params counts *all* parameters required for inference, including RoI-proposals, the architecture, and output heads (see §G).

| | Model | Top-1 Base | Top-1 Large |
|---|---|---|---|
| *Supervised* | ViT-B/16 (Dosovitskiy et al., 2021) | 77.9 | 76.5 |
| | DeiT$_{\text{Base}}$ (Touvron et al., 2021) | 83.1 | - |
| | Swin$_{\text{Base}}$ (Liu et al., 2021) | **84.5** | - |
| *Self-supervised* | DINO (Caron et al., 2021) | 82.8 | - |
| | MoCo v3 (Chen et al., 2021) | 83.2 | 84.1 |
| | MaskFeat (Wei et al., 2021) | 83.6 | 85.7 |
| | SimMIM (Xie et al., 2022) | 83.8 | 85.4 |
| | BEiT* (Bao et al., 2022) | **84.6** | 85.2 |
| | MAE (He et al., 2022) | 83.6 | 85.9 |
| | **VLC** (ours) | **84.5** | **86.3** |

Table 5: Our pretraining does not compromise the quality of image representations, as evidenced by our superior results on ImageNet classification. *Supervised* models are pretrained on ImageNet 1K and *self-supervised* models are evaluated by end-to-end fine-tuning. *BEiT uses a DALLE pretrained tokenizer.

**Better Grounding translates to compositional reasoning** We find that the finetuned patch-text affinities translate to greater compositional reasoning performance. We report inference-only results on Winoground (Thrush et al., 2022) with our Refcoco-finetuned model in Table 7. Winoground requires pairing up two sets of images and sentences with minimally contrastive semantics. Instead of directly predicting a pairing logit using the image-text matching head, it is more effective to measure image-sentence association via grounding success conditioned on an input sentence. Concretely, by treating the max affinity as the image-sentence-matching score, our Refcoco-finetuned model achieves a state-of-the-art Group Score and outstanding Image Scores on Winoground. This superior reasoning performance again verifies that our pretrained representations are more capable of using text/images to disambiguate each other.

| Model | Text | Image | Group |
|---|---|---|---|
| MTurk Human | 89.50 | 88.50 | 85.50 |
| Random Chance | 25.00 | 25.00 | 16.67 |
| UNITER$_{\text{Base}}$ | 32.25 | 13.25 | 10.00 |
| VILLA$_{\text{Base}}$ | 30.00 | 12.00 | 8.00 |
| ViLT$_{\text{Base}}$ | 34.75 | 14.00 | 9.25 |
| CLIP$_{\text{Base}}$ | 30.75 | 10.50 | 8.00 |
| FLAVA-ITM$_{\text{Base}}$ | 32.25 | **20.50** | 14.25 |
| **VLC**$_{\text{Base}}$ | 28.00 | 19.75 | 12.50 |
| UNITER$_{\text{Large}}$ | **38.00** | 14.00 | 10.50 |
| VILLA$_{\text{Large}}$ | 37.00 | 13.25 | 11.00 |
| **VLC**$_{\text{Large}}$ | 32.00 | 20.00 | **14.75** |

Table 7: Winoground is a challenging test-only set for visio-linguistic compositional reasoning. **VLC**$_{\text{Large}}$ is competitive among similar-sized models without a second-stage pretraining.

### 5.3 Ablations

**Pretraining Objective** We ablate different combinations of objectives and train **VLC**$_{\text{Base}}$ with 4M image-text pairs. Figure 4 (left) shows that, as the training steps increase, there is a consistent improvement for **VLC** with MIM. This contrasts to findings in previous work (Dou et al., 2022a; Kim et al., 2021; Li et al., 2021a). We compare **VLC**$_{\text{Base}}$ with different initializations and the effects of unimodal pretraining in Appendix E.

**Downstream Adaptation** Since we finetune for 30 epochs on the grounding task vs. 10 epochs on the retrieval or understanding tasks, we would like to understand whether the grounding capability is simply a result of intensive finetuning. To this end, we experiment with our **VLC** checkpoint as well as other three pretrained checkpoints: ViT (Dosovitskiy et al., 2021), ViLT (Kim et al., 2021) and MAE (He et al., 2022).

Please refer to the Winoground (Thrush et al., 2022) paper for how their evaluation metrics are defined.

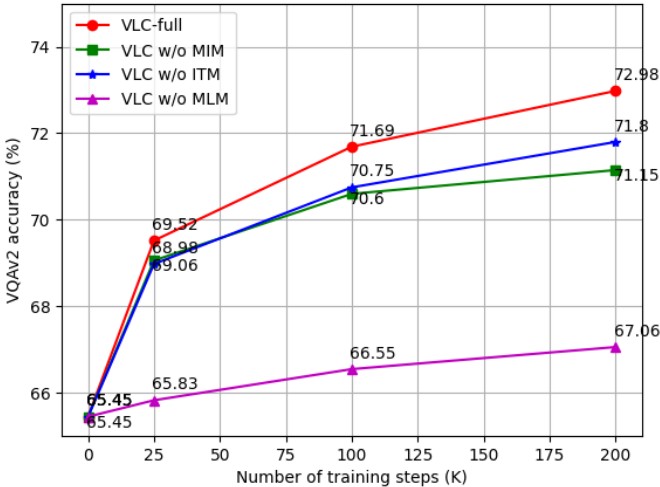 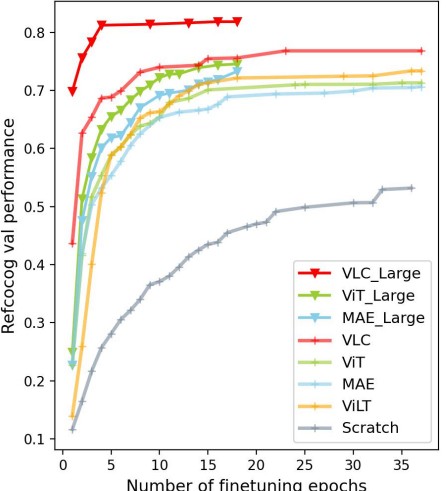

Figure 4: **Left:** Role of objective functions. Our experiments show that $\mathbf{VLC}_{\text{Base}}$ sees gains from including MIM and that increased training steps continues to improve VQAv2 performance. **Right:** Comparing the effects of different pretrained V+L checkpoints on downstream adaptation. Our experiments show that the pretrained **VLC** representations enable faster finetuning and better downstream performance.

We follow the same finetuning and inference procedures for all checkpoints. Figure 4 (right) plots validation accuracy curves against the number of finetuning epochs. **VLC** exhibits a considerable advantage right from the beginning, and maintains a clear margin until convergence. The relative advantage of **VLC** becomes even greater when we compare the *Large* model variants. This result validates that our proposed pretraining scheme provides a better foundation for finetuning, easing the computation burden of downstream adaptation.

**Initialization+Scaling** In Figure 5, we compare VQAv2 test-dev accuracy between **VLC** with three baselines that use pretrained object detectors and BERT for initialization. We see that while supervised initialization provides strong priors for the VQA task (*e.g.*, VinVL), scaling the model size only has a marginal improvement. As a comparison, our model is initialized with MAE pretrained on ImageNet1K without labels. There is a substantial gain when scaling to a larger model.

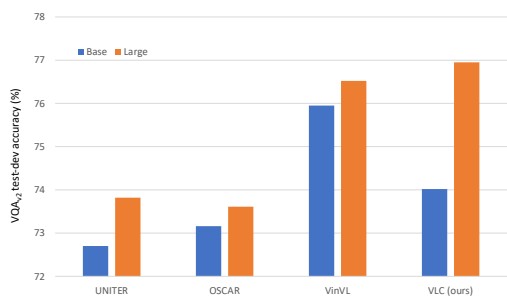

Figure 5: Comparison of the benefit of model scaling. While models initialized with supervised checkpoints inherit strong priors, **VLC** exhibits the most substantial improvement when scaling the model size.

# 6 Understanding the models

**VLC** is designed in importantly different ways from traditional bounding-box based approaches. In particular, while the visual stack is traditionally frozen in those models, now the entire "backbone" is learnable. Also, where previously, the goal was to "map" vision to language, now the two are learned jointly. We therefore take this opportunity to investigate the models to better understand how their behaviors differ due to the two key design choices. For a fair comparison with ViLT, we use $\mathbf{VLC}_{\text{Base}}$ which is trained with the same model architecture and image-text pairs.

**Understanding Patches.** We begin with a simple patch clustering visualization (Figure 6). Without the inclusion of any language, we can simply cluster (and color) the visual patch embeddings of ViLT and **VLC**. ViLT relies on on larger patches ($32 \times 32$) for higher resolution ($384 \times 640$). We instead use smaller patches and lower resolution ($16 \times 16$ for $384 \times 384$). It is easy to see how both models are identifying key semantic

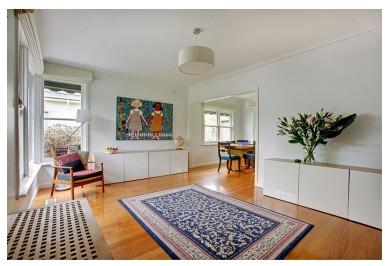 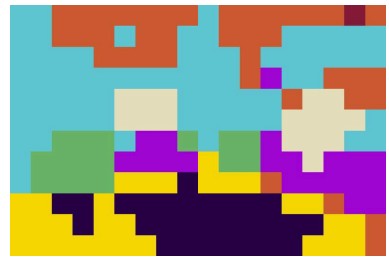 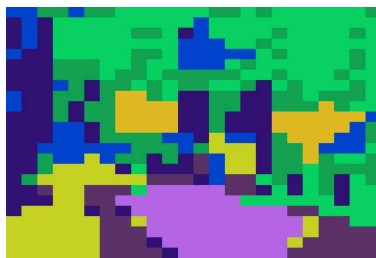

Original Image                ViLT clusters                **VLC** clusters

Figure 6: Visualization of patch clusters for an example image as produced from ViLT (many densely clustered patches) versus **VLC**'s more fine-grained and diffuse representations. We believe this representational difference makes for easier and faster learning and scaling – akin to "fast mapping" in language acquisition.

regions of the image (e.g. the rug, painting and plant). Also note, both models incorrectly place the painting and plant in the same cluster.

We leverage the nocaps dataset (Agrawal et al., 2019) to investigate representation diversity at scale. Nocaps provides captions for images from three buckets: inside COCO (in-domain), similar to COCO (near-domain), and outside COCO (out-domain). We compute representation similarities between patches and nouns for these three image buckets separately, and seek to answer the following questions: (1) Are ViLT patches more tightly clustered – perhaps due to the discriminative training objective, and (2) How do both models' representation diversity change for images more (or less) like ImageNet images. In Figure 7, we see several trends. First, ViLT's "most similar" patch to the noun rarely has a similarity greater than 0.1, indicating that information initially coming from two modalities are not well integrated into each other's representations. Second, we see the mass of patch-noun similarity distributions for ViLT slightly shift towards zero as we move from in-domain to out-domain, indicating that ViLT's difficulty in finding alignments to novel words. **VLC** has a significantly smoother distribution of patch-noun similarities. **VLC** often represents a nontrivial set of patches largely close to nouns across all image buckets. Also, the centers of similarity mass do not shift as much as those in ViLT plots as we move to out-domain images. A caveat is that these plots do *not* justify the semantic meaningfulness of the alignment, but they do show starkly different behaviors between ViLT and **VLC**. Our findings are supported by more visualized examples in F.

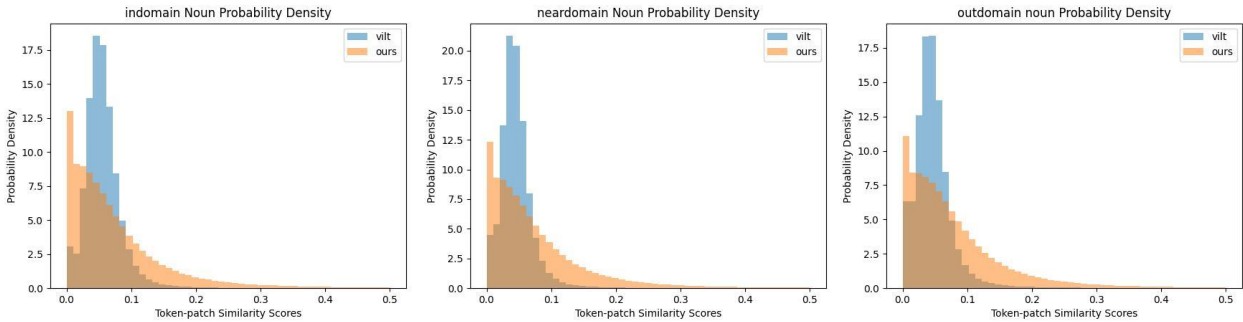

Figure 7: We plot the distribution of patch-noun similarities to compare the representation quality between ViLT and **VLC**$_{\text{Base}}$. ViLT rarely represents patches in great proximity to nouns, likely due to its discriminative pretraining objective. ViLT also tend do produce weaker alignment for images dissimilar to ImageNet images. In contrast, **VLC**$_{\text{Base}}$ produces patch representations strongly-aligned to corresponding nouns and does not show a preference towards ImageNet images.

**Image Classification.** Given that the underlying visual representations are shifting through the cross-modal training, we run a simple image classification experiment to see the effects language training has on the underlying visual "backbone". We compare **VLC** with state-of-the-art models on ImageNet-1K classification and report top-1 validation accuracy of a single $384 \times 384$ crop.

The bird is on the branch with leaves alone

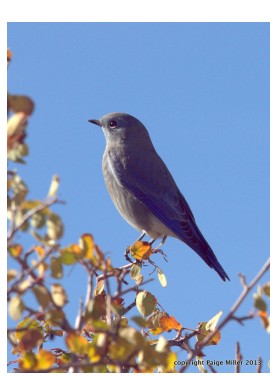 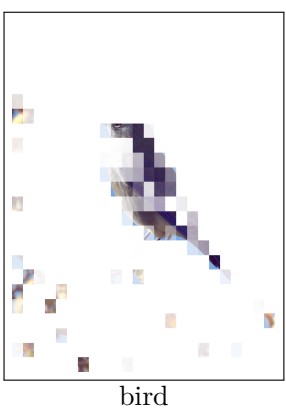 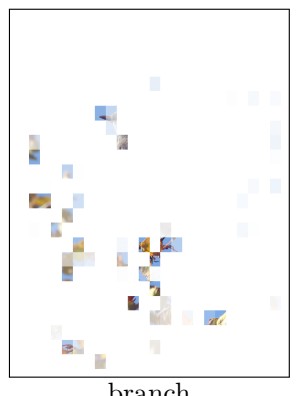 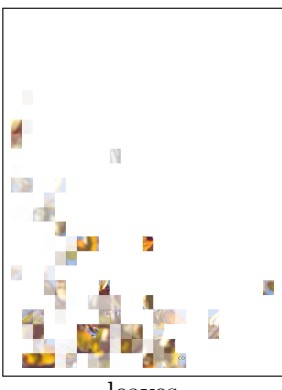

bird  branch  leaves

Figure 8: Finer-grained patch-token alignments is implicitly learned from weakly-aligned pretraining data. We visualize patch-similarities with different tokens from the same caption to see how our model uniquely represents them. It is interesting to note that our model attempts to differentiate branches from leaves.

As shown in Table 5, **VLC** learns generic representations which are transferable to ImageNet classification. After finetuning on ImageNet-1K, our model matches the performance of $Swin_{Base}$ (Liu et al., 2021) that is trained with supervised labels. Note that BEiT (Bao et al., 2022) is a two-stage pretrained model of which the image tokenizer is trained on 250M examples of DALLE (Ramesh et al., 2021) data. Compared with MAE (He et al., 2022), our model learns competitive multi-modal representations from vision-language pretraining while retains high-quality image representations.

**Evaluation on Open-domain VQA.** To investigate if the alignments between image patches and text tokens are semantically meaningful, we evaluate our **VLC** on A-OKVQA dataset (Schwenk et al., 2022). A-OKVQA is more demanding than VQA (Goyal et al., 2017) for it requires commonsense reasoning about the scene depicted in the image. In *multiple choice* (MC) setting, a model chooses its answer from one of four options. In the *direct answer* (DA) setting, a model can generate any text as its answer, which better applies to real-world scenarios. We use **VLC** as the mulitmodal encoder and a pretrained BERT to generate answers. In Table 8, we compare **VLC** with large-scale pretrained **discriminative** models - BERT (Devlin et al., 2019) and ResNet (He et al., 2016), **contrastive** model - CLIP (Radford et al., 2021), **generative** model - GPT3 (Brown et al., 2020), and models **specialized** on open-domain VQA. While CLIP stands out in the MC setting, it performs worse than other baselines in the DA setting. KRISP (Marino et al., 2021)

| Model | Model Size | MC-test | DA-test |
|---|---|---|---|
| Random | 0 | 25.36 | 0.06 |
| Large-scale Pretrained model | | | |
| BERT (Devlin et al., 2019) | 110M | 33.54 | 8.41 |
| GPT-3 (Brown et al., 2020) | 175B | 35.21 | 11.49 |
| ResNet (He et al., 2016) | 23M | 28.81 | 2.30 |
| CLIP (Radford et al., 2021) | 150M | **51.01** | 7.10 |
| Specialized model | | | |
| ViLBERT (Lu et al., 2019) | 274M | 41.5 | 25.9 |
| LXMERT (Tan & Bansal, 2019) | 240M | 41.6 | 25.9 |
| KRISP (Marino et al., 2021) | 300M | 42.2 | 27.1 |
| **VLC**$_{Base}$ (ours) | 195M | 44.82 | **27.49** |
| ClipCap[†] (Mokady et al., 2021) | 930M | 51.43 | 25.90 |
| GPV-2[‡] (Kamath et al., 2022) | 380M | 53.7 | 40.7 |

Table 8: A-OKVQA under both Multiple-Choice (MC) and Direct Answer (DA) settings. [†]ClipCap uses pretrained CLIP+GPT2$_{Large}$. [‡]GPV2 uses pretrained VinVL+T5$_{Base}$ and learns from Bing data. We report accuracy (%) returned from the evaluation server.

ensembles different pretrained image classification and object detection models to exact image features. As a comparison, our **VLC** outperforms KRISP in both settings. This implies that our model provides richer and language-aware image features which are conducive to open-vocabulary text generation. Note that ClipCap (Mokady et al., 2021) and GPV2 (Kamath et al., 2022) use either much more data for pretraining or finetuning on the open-domain VQA task.

# 7 Visualizations

Patch-based transformer architectures allow for intuitive visualizations of the lexical alignment. We first demonstrate that faithful alignment corresponding to entity names or attributes already exist in the pretrained representations. Note, there is no explicit supervision on fine-grained patch-token alignment, yet the model learns to do so from weakly-aligned data. In Figure 8, we show results from visualizing three different words in the same caption for an image from COCO. Note that for the word branch, the model is actively attempting to avoid the abundant leaves. Second, since there is nothing about our model besides the MAE initialization that should be biased (as shown previously) towards ImageNet classes, we present three images in Figure 9 that highlight words not present in the standard ImageNet1K training split used by other models. Specifically, a noun (*string*), adjective (*yellow*), and verb (*swinging*). These demonstrate the general trend of ViLT often focusing on surprising locations.

Moreover, Section 5.2.3 shows that finetuning can further incentivize the affinity patterns to behave like bounding boxes, which is the standard output format for localization. We visualize predicted alignment after finetuning in Table. 9. We highlight a greater reasoning ability beyond recognition in terms of disambiguating visual entities based on how they are referenced linguistically.

Caption with focus

| Original Image | ViLT | **VLC** |

A person on a beach holding a kite string and a kite is in the air

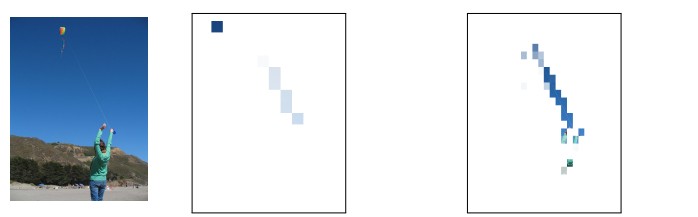

A cat sitting on a chair, that is blue and yellow

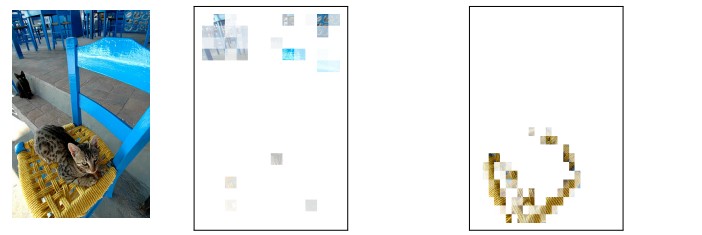

A baseball player swinging a baseball bat at a baseball

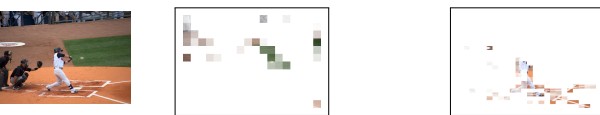

Figure 9: Out-of-domain concepts and images (i.e. not in ImageNet-1K). We visualize patch-similarities with a noun (top), an adjective (middle) and a verb (bottom), respectively. The model delicately avoids nearby but distinct concepts (e.g. the cat on the chair or irrelevant parts of the baseball field). More examples and analysis can be found in Appendix F.

# 8 Conclusion

We present **V**ision-**L**anguage from **C**aptions (**VLC**), a generic vision-language model pretrained with *only* image-caption pairs. It uses a single linear layer to project raw pixel stimuli and token embeddings into the same representation space, followed by Transformer blocks jointly modeling two modalities. By removing the dependency on image region proposals, our model is both (1) more data efficient, for it does not require pretraining-scale class labels or bounding box annotation, and (2) faster at inference, for it does not require a tedious vision-only branch.

Despite being lighter and faster, **VLC** performs competitively on a diverse set of vision-language tasks, as compared to existing approaches relying on detection or ImageNet supervision. The MIM pretraining objective encourages richer and language-aware visual representations, which implicitly results in finer-grained alignment between patch and token representations. With moderate downstream finetuning, **VLC** can be easily adapted to (1) perform multi-modal retrieval, (2) answer questions, (3) reason about visual information guided by free-form language, or (4) ground linguistically-referenced objects into bounding boxes. **VLC**'s strong performance across nine downstream benchmarks clearly demonstrate the wide task applicability of a unified vision-language encoder. Our results appeal to the community to rethink the general multimodal pretraining pipeline and to reduce unnecessary cost of pretraining unimodal encoders. As performance of

| Input image | Predicted affinity map | Predicted bbox reduced | Predicted & Ground-truth |
|---|---|---|---|
| 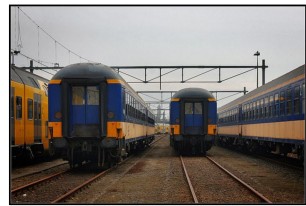 | 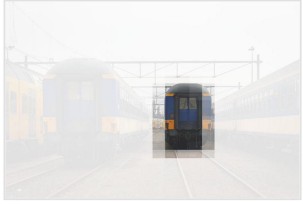 | 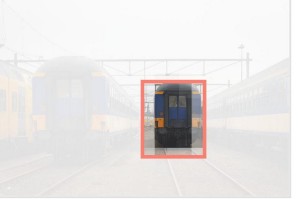 | 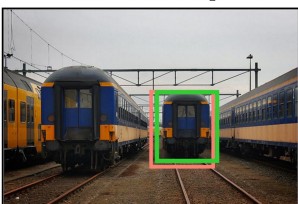 |

**the leftmost train** — The model succeeded at distinguishing same-type instances via absolute position.

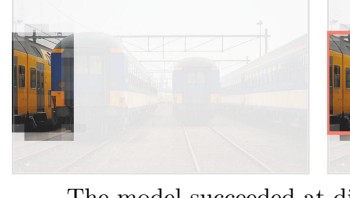

**the smallest train** — The model succeeded at distinguishing same-type instances via absolute size.

Table 9: Patch-token affinities can be finetuned towards localizing objects specified by referring expressions.

**VLC** scales with increased training data, this opens an exciting avenue for large-scale weakly-supervised open-domain vision-language models.

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

# Appendix

## Table of Contents

## Appendix

This supplementary material has seven sections. Section A describes the details of our pretraining datasets. Section D.3 measures the inference time of different models and highlights the efficiency of our model. Section C and D describe implementation details for downstream tasks. Section F shows more visualization examples with more comparisons. Section G includes additional results and error analysis of **VLC** finetuned on grounding tasks. Section H discusses the limitations and societal impact of this work.

## A  Pre-training dataset

The statistics of the pretraining dataset is shown in Table A4. Most of the existing approaches, such as UNITER (Chen et al., 2020b) and ViLT (Kim et al., 2021), use MSCOCO, VG, GCC and SBU to pre-train their models. We denote this training set as *base*. To verify the scalability of our model, we follow VinVL (Zhang et al., 2021) to further incorporate VQA, VG-QA, GQA, Flickr30K and OpenImages. As there are some overlaps among VG, MSCOCO and VQA, we exclude all those training images that appear in the downstream tasks via URL matching.

Our extended VLC-for-grounding pipeline also strikes a superior efficiency-performance balance, with visualizations shown in Figure A1 where efficiency is indicate by both inference time and parameter count.

Inference time is reported on 1 RTX2080Ti with batch_size=1, averaged across the same 1K randomly selected samples from Refcocog_val(umd) over 5 launches. Since different approaches require different post-processing for reducing an output Tensor to bbox coordinates, we time execution starting from feeding raw image and text tensors into the pipeline until the four-item coordinates tuple is obtained. Bespoke grounding models under comparison often involve components overspecialized for the end-goal of coordinate prediction, at the expense of directly optimizing for well-aligned V+L representations. Our results reveal that centralizing the expressive power in a single cross-modal encoder benefits V+L unification and efficiency without loss in performance.

# B    Additional experimental results

In our experiments, we carefully regulate the quantity of pretraining data utilized for two key aspects: the initialization checkpoint, which encompasses sources such as ImageNet (Russakovsky et al., 2015), COCO (Lin et al., 2014), WebLI (Chen et al., 2023), and the vision-language pretraining phase. It is essential to underscore that numerous works on vision-language pretraining tend to downplay the significance of data and computation resources already consumed by their unimodal initializations. Consequently, this inadvertent oversight conceals the fact that their ultimate outcomes necessitate substantially greater resources than would be equitable for a meaningful comparison. In this section, We present supplementary comparisons that demonstrate our model can achieve better or competitive results on ImageNet-1K and GLUE, while utilizing substantially smaller amounts of data. The results implies that our model is friendlier to downstream applications after balancing resource demands and representation quality.

| Model | #Params | #Images |
|---|---|---|
| PaLI (Chen et al., 2023) | 3B-17B | 1B |
| Flamingo (Alayrac et al., 2022) | 3.2B-80B | 1.8B |
| CoCa (Yu et al., 2022) | 383M-2.1B | 4.8B |
| CLIPPO (Tschannen et al., 2023) | 316M | 10B |
| SimVLM (Wang et al., 2022c) | 300M-1B | 1.8B |
| VLC (ours) | 307M | 5.6M |

Table A1: Recent works employ larger training sets and model sizes to emphasize the final performance.

## B.1    Natural Language Understanding

We finetune our model on GLUE benchmark. Despite having non of the task-specific supervision and being pretrained on less text, our model still performs competitively on pure linguistic tasks.

| Tasks | CoLA | QQP | QNLI | SST-2 | STS-B | MRPC | RTE | MNLI |
|---|---|---|---|---|---|---|---|---|
| SimVLM (Wang et al., 2022c) | 46.7 | **90.4** | **88.6** | 90.9 | - | 84.4 | **63.9** | **83.4** |
| CLIP (Tschannen et al., 2023) | 6.6 | 82.7 | 73.0 | 86.2 | 65.0 | 81.4 | 53.8 | 71.8 |
| CLIPPO (Tschannen et al., 2023) | **55.3** | 87.9 | 86.7 | **94.2** | 85.8 | 85.9 | 59.2 | 82.3 |
| VLC (ours) | 41.8 | 89.7 | 87.0 | 89.7 | **87.0** | **88.6** | 59.6 | 78.8 |

Table A2: Our **VLC** performs competitively on the GLUE benchmark despite that it is pretrained on less text without task-specific supervision.

## B.2    Image Classification

We present a comparative analysis of performance in the ImageNet-1K image classification task. Despite utilizing a substantially reduced image dataset, our **VLC** surpasses the achievements of existing works, thus demonstrating the efficacy of our model. We adhere to the hyperparameters specified in MAE (He et al., 2022) without any additional tuning.

| Model | #Params | #Images | ImageNet-1K (finetune) |
|---|---|---|---|
| CLIP (Mu et al., 2022) | 350M | 15M | 81.0 |
| SLIP (Mu et al., 2022) | 350M | 15M | 84.2 |
| CoCa (Yu et al., 2022) | 393M | 3B | 84.9 |
| VLC (ours) | 307M | 5.6M | **86.3** |

Table A3: Our **VLC** demonstrates superior performance in the ImageNet-1K image classification task, despite employing significantly smaller image data.

| Dataset | MSCOCO | VG | GCC | SBU | VGA | GQA | VG-QA | Flickr30K | OpenImages |
|---|---|---|---|---|---|---|---|---|---|
| # Images | 113K | 100K | 2.95M | 860K | 83K | 79K | 87K | 29K | 1.67M |
| # Text | 567K | 769K | 2.95M | 860K | 545K | 1026K | 931K | 145K | 1.67M |

Table A4: Statistics of the pre-training dataset

## C   Implementation Details for Downstream Understanding & Retrieval Tasks

For downstream understanding and retrieval tasks, we fine-tune our model with a learning rate of $5e^{-4}$ for 10 epochs. We use $480 \times 480$ as the input image resolution for the VQA task and $384 \times 384$ for NLVR$^2$ and image-text retrieval tasks.

**Visual Question Answering (VQA (Goyal et al., 2017)).**  Given an input image and a question, the VQA task is to predict an answer from the visual content. We conduct experiments on VQAv2 dataset (Goyal et al., 2017) that is built on MSCOCO. It contains 83K images for training, 41K for validation, and 81K for testing. Following previous work (Tan & Bansal, 2019; Chen et al., 2020b; Li et al., 2021a), we use the training, validation splits and additional question-answer pairs from Visual Genome while reserving $1,000$ validation image-question pairs for internal validation. We report performance on the test-dev and test-std splits. We follow the standard practice (Kim et al., 2021) to convert the task to a multilabel classification task with $3,192$ answer classes.

**Natural Language for Visual Reasoning (NLVR$^2$ (Suhr et al., 2018)).**  Given a triplet of two images and a description, this task is to predict whether this description describes a pair of images. Following previous work (Kim et al., 2021; Chen et al., 2020b), we use the *pair* method which treats one input sample as two image-text pairs by repeating the text twice. Each pair is passed through our model and we take the concatenation of two pooled representation [CLS] from our model as the representation of one input sample. Similar to the settings of the VQA task, we use a 2-layer MLP with a hidden size of $1,536$ to adapt **VLC** to the NLVR$^2$ task.

**Image-Text Retrieval.**  Image-Text retrieval contains two subtasks: image-to-text retrieval (TR) and text-to-image retrieval (IR). We evaluate our pre-trained models on the Karpathy splits (Karpathy & Fei-Fei, 2015) of MSCOCO (Lin et al., 2014) and Flickr30K (Plummer et al., 2015) in fine-tuning settings. MSCOCO contains 123K images, and each image has five corresponding human-written captions. We split the data into 82K/5K/5K training/validation/test images. To be consistent with previous work (Kim et al., 2021; Chen et al., 2020b), we use the additional 30K images from MSCOCO validation set to improve the performance. Flickr30K contains 31K images with five captions for each image. We split the data into 30K/1K/1K as the training/validation/test set.

## D   Implementation Details for Downstream Grounding Tasks

The main text has briefly argued that appropriate supervision enables generalizable and versatile representations which naturally translate to downstream grounding performance without tailored design. This section describes our approach of adapting the Transformer backbone to performing grounding tasks in detail. We conduct grounding experiments on Refcoco/+/g. Namely, Section D.1 provides dataset details, Section D.2

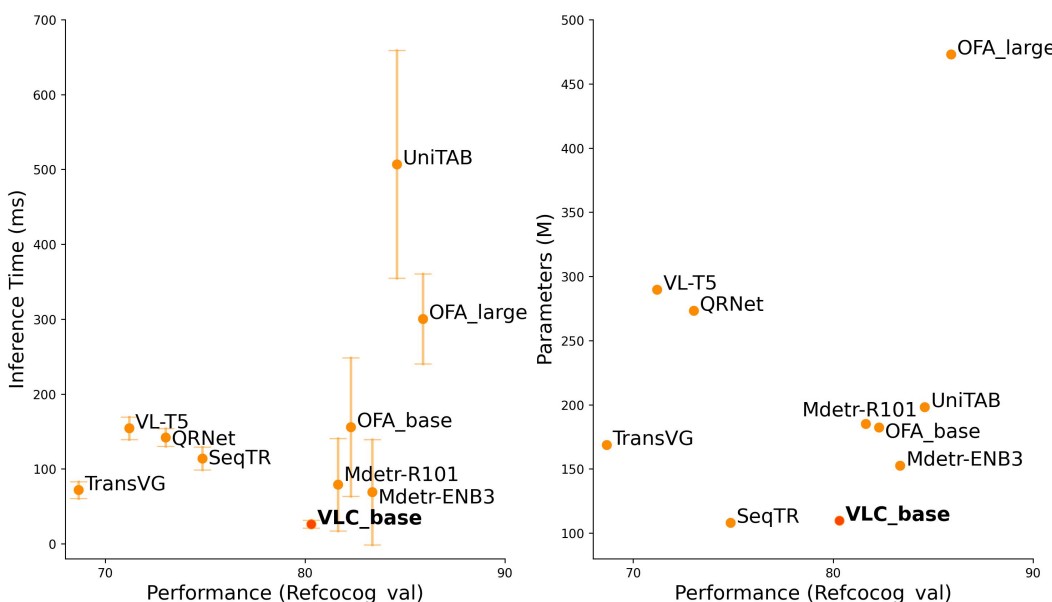

Figure A1: Inference time and parameter counts of different Referring Expression models. Our **VLC**$_{\text{Base}}$ achieves competitive performance with considerably less inference time (left) and parameter count (right).

introduces the process of converting bounding box annotations to patch regions for supervised finetuning, and Section D.3 describes how the resulting affinities can be re-interpreted as either a mask or a bounding box.

### D.1 Referring Expression Datasets

The referring expression task feeds a model with an (image, region description) pair, then expects a localization output as either a bounding box (bbox) or a pixel-level mask corresponding to the description. It tests for visual grounding and reasoning as the model indicates which part of the image is referenced by the text. Note, a mask output is more challenging to produce since a mask can adopt various shapes and requires a dense prediction in the original image's resolution.

We use three Referring Expression datasets: Refcoco, Refcoco+, and Refcocog. All of which use MSCOCO (Lin et al., 2014) training images. Refcoco and Refcoco+ were collected from human annotators participating in a "ReferIt" Game, where one player selected regions in an image referenced by a text description provided by the other player. Where they differ is that Refcoco+ disallowed players from using location-based expressions (e.g. "person to the right"), thereby being purely appearance-based. Refcocog was collected in a non-game setting but contains more linguistically complex descriptions.

Bounding boxes are evaluated by Intersection-over-Union (IoU), while masks are evaluated by pixel-level IoU. To compare with other relevant models, we follow the literature by reporting IoU_acc, which is the proportion of testing samples on which the model achieves an IoU > 0.5.

### D.2 Finetuning

Our approach builds on the notion of an affinity map. Formally, we define the affinity map, $\mathcal{A}$, as the cosine distance between all $\mathcal{L}$anguage tokens and $\mathcal{V}$isual patches.

$$\hat{\mathcal{A}}_{t,p} = \cos(\mathcal{L}_t, \mathcal{V}_p) \tag{7}$$

for $T$ tokens and $P$ patches. This resulting matrix $[-1, 1]^{T \times P}$ provides a normalized score for the relationship between every token and patch.

**Supervision Signals**  For supervision, we translate bounding box annotations to sets of patches and phrases simply correspond to their indices, with `[CLS]` being the initial index 0. The ground-truth affinity scores can be obtained from labeled datasets where annotators selected bboxes corresponding to phrases in an (image, sentence) pair. Thus, for an (image, sentence) pair, we have a set of annotated (bbox, phrase) pairs, from which we would like to get a $T \times P$ matrix of affinity scores. Note that the full image has dimensions $H \times W$ and will be divided into a $\sqrt{P} \times \sqrt{P}$ grid.

For a given annotated (bbox, phrase) pair, if the lexical token $\mathcal{L}_t$ belongs to the phrase and patch $\mathcal{V}_p$ overlaps with the bbox, we set $\mathcal{A}_{t,p}$ in the ground-truth affinity map to be amount the patch overlaps with the bbox.

$$\mathcal{A}_{t,p} = \frac{|Patch \cap \mathrm{BBox}|}{|Patch|} \tag{8}$$

If a bbox corresponds to the entire sentence, we set ground-truth affinity scores for the 0-th token, `[CLS]`.

**Hyperparameters**  We opt for a higher resolution (384) and smaller patches (16) since the patch size will determine the granularity of our predictions. This yields a final 24×24 grid. We finetune VLC on the combined Refcoco/+/g training sets for 50 epochs with AdamW (Loshchilov & Hutter, 2018) optimizer and a $5e^{-4}$ learning rate.

**Finetuning Objective**  We optimize our model with a binary-cross-entropy (BCE) loss between the ground-truth and predicted affinity scores.

$$Loss = \sum_{t \in T, p \in P} \mathrm{BCE}(\mathcal{A}_{t,p}, \hat{\mathcal{A}}_{t,p}) \tag{9}$$

We intentionally avoided the use of a softmax because normalization would force competition across patches and affect their individual judgments. Each token-patch pair relationship is independent during loss calculation.

Note, that the domain expected by BCE is the full real line, but our affinities are computed via cosine similarities and are therefore bounded to [-1,1]. Empirically, we found that re-scaling the input by $k = 2$ improved performance due to a better utilization of the [0,1] output range. However, performance decreases with a larger k because it tends to over-sharpen the sigmoid's decision boundary, leaving less "wiggle room" for the raw logits.

## D.3  Inference

The forward pass through the Transformer outputs a single tensor of shape $L \times 768$, or more accurately, the concatenation of two tensors of shape $T \times 768$ and $P \times 768$ for the encoded $\mathcal{L}$anguage and $\mathcal{V}$isual sequences, respectively. During inference, these matrices are used in Equation 7 to compute the predicted affinity matrix $\hat{\mathcal{A}}$. Next, we describe how a mask or a bounding box can be algorithmically derived from a predicted affinity matrix $\hat{\mathcal{A}}$. We emphasize that this procedure requires no additional learned parameters and is algorithmically efficient.

Since rows in $\hat{\mathcal{A}}_{actv}$ associated with a phrase can be pooled into a single row and reshaped into $\sqrt{P} \times \sqrt{P}$, we henceforth refer $\hat{\mathcal{A}}_{actv}^r$ to such a $[-1, 1]^{\sqrt{P} \times \sqrt{P}}$ matrix with spatial correspondence to the image, albeit with a coarser resolution. Then, a binary mask can be simply derived by bilinearly interpolating $\hat{\mathcal{A}}_{actv}^r$ back to the original resolution, and binarizing the values with a threshold.

The procedure for deriving a bbox from $\hat{\mathcal{A}}_{actv}^r$ is more complicated, which we detail below. **We formulate bbox prediction as a search problem**. A bbox is denoted by $\mathcal{B} = (x, \ y, \ w, \ h)$, satisfying $0 \leq x \leq x + w \leq W$, $0 \leq y \leq y + h \leq H$. The goal is to search for $\mathcal{B}$ given $\hat{\mathcal{A}}_{actv}^r$, such that $\mathcal{B}$ best represents a rectangular region $\hat{\mathcal{A}}_{actv}^r$ tries to highlight (Figure A2, Left). To solve the search problem we need to decide on the following: the search space, the search order, and the search criterion.

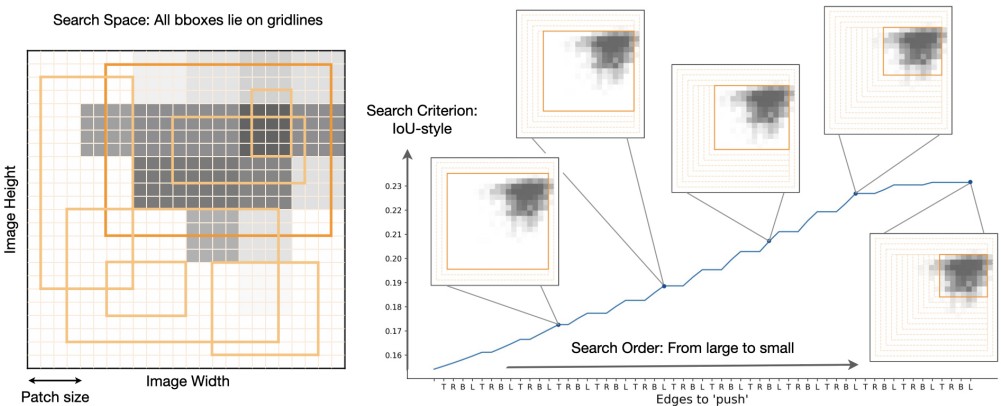

Figure A2: Inference procedure overview for "Affinity Map ⇒ BBox", which is formulated as a search problem. Left: we define the search space to contain all candidate bboxes that lie on gridlines. Right: our algorithm greedily iterates over bboxes following the search order of large → small, and terminates once the search criterion no longer improves.

**Search space** Theoretically, we have an infinite number of boxes to search over which can be as large as the input image or as small as nearly a single point. We constrain our search to only those boxes whose edges lie on gridlines, with step $t$ denoting the unit between two gridlines.

**Search order** We start with a box encompassing the full image and greedily search for progressively smaller boxes.

**Search criterion** We propose a satisfying criterion $\mathcal{M}$ that guarantees equivalence between $\hat{\mathcal{A}}^r_{actv}$ and the theoretically optimal bbox ($\mathcal{B}$). In the case where all values in $\hat{\mathcal{A}}^r_{actv}$ are binary, this reduces to calculating a standard IoU (*Intersection-over-Union*). In practice, values will be real-valued. We overload the notation to handle the generalization from binary to real values in $\hat{\mathcal{A}}^r_{actv}$. Intersection is defined as the sum of the active values within $\mathcal{B}$. Union is equal to the sum of three terms, namely the active values inside $\mathcal{B}$, the values that should have been active inside $\mathcal{B}$ and the active values outside $\mathcal{B}$. The first two terms in Union add up to $|\mathcal{B}|$. As a single equation this results in

$$\mathcal{M} = \frac{|\hat{\mathcal{A}}^r_{actv} \cap \mathcal{B}|}{|B| + |\hat{\mathcal{A}}^r_{actv} \cap \neg\mathcal{B}|} \tag{10}$$

In practice, this is an easy calculation, as the intersections and negations simply correspond to summing the values inside and outside of the candidate box. Further, summing (in lieu of counting) trivially generalizes to the continuous valued activations.

We provide additional proof of the equivalence to optimizing F1 in the supplementary. We will note, there is a potential hyperparameter which we will call $C$ that defines the desired "tightness" of the box.

Next we leverage the the search space, order and criterion ($\mathcal{M}$) in Algorithm 1 to predict bounding boxes. Initializing $\mathcal{B}$ to surround the entire image, our *PUSH* algorithm greedily iterates over progressively smaller $\mathcal{B}$ and terminates when $\mathcal{M}(\hat{\mathcal{A}}^r_{actv}, \mathcal{B})$ no longer improves. Within each iteration, there are "push" attempts on the edges of $\mathcal{B}$, one at a time, by $t$-amount inwards according to an order $\mathcal{O} \in Permutation([T, L, B, R])$. For example, $\mathcal{O} = [T, R, B, L]$ would translate to a clockwise progression of tightening edges from the $Top - Right - Bottom - Left$. A push operation is successful if and only if it increases $\mathcal{M}$. Otherwise, the edge is left unchanged. The algorithm terminates when it encounters unsuccessful push attempts on all four edges in a row. Figure A2 visualizes a *PUSH* execution.

---

For example, $C = 0.5 * |\hat{\mathcal{A}}^r_{actv}|$ would produce bboxes systematically smaller than $C = |\hat{\mathcal{A}}^r_{actv}|$. The supplementary discusses about where $C$ is derived from. In this work, we leave the scaling to 1 and allow future work to analyze how $C$ would influence the returned value of Algorithm.1

---

**Algorithm 1:** PUSH

---

**input** : heatmap $\hat{\mathcal{A}}^r_{actv}$, image size $(W, H)$, step $t$, measure $\mathcal{M} : (\hat{\mathcal{A}}^r_{actv}, \mathcal{B}) \to \mathcal{R}$,
$\quad\quad$ order $O \in Permutation([T, L, B, R])$

**output**: $\mathcal{B}$: box coordinates $(x, y, w, h)$ $s.t.$ $0 \leq x \leq x + w \leq W$ and $0 \leq y \leq y + h \leq H$

---

$(x, y, w, h) \leftarrow (0, 0, W, H)$
$\mathcal{B} \leftarrow (x, y, w, h)$
**while** $w > 0$ *and* $h > 0$ *and* $move \neq False$ **do**
$\quad$ **for** $e \in O$ **do**
$\quad\quad$ **if** $e == T$ **then** $\mathcal{B}' \leftarrow (x, y + t, w, h - t)$
$\quad\quad$ **else if** $e == B$ **then** $\mathcal{B}' \leftarrow (x, y, w, h - t)$
$\quad\quad$ **else if** $e == R$ **then** $\mathcal{B}' \leftarrow (x, y, w - t, h)$
$\quad\quad$ **else** $\mathcal{B}' \leftarrow (x + t, y, w - t, h)$

$\quad\quad$ **if** $\mathcal{M}(\hat{\mathcal{A}}^r_{actv}, \mathcal{B}') > \mathcal{M}(\hat{\mathcal{A}}^r_{actv}, \mathcal{B})$ **then**
$\quad\quad\quad$ $\mathcal{B} \leftarrow \mathcal{B}'$
$\quad\quad\quad$ $(x, y, w, h) \leftarrow \mathcal{B}$
$\quad\quad\quad$ $move \leftarrow True$
$\quad\quad$ **end**
$\quad$ **end**
**end**
**return** $\mathcal{B}$

---

In our experiments, we use values $P = 24$, $W = H = 384$ and $t = \frac{1}{4} * patch\_size = 4$. A known limitation of our approach is highly discontiguous or concave affinity patterns. We did not observe this condition in our setting. We provide empirical evidence that our "Affinity Map $\Rightarrow$ BBox" formalization is effective but leave a mathematically rigorous investigation to future work.

### D.4 Additional Math Proof

In the section above, we derive the search criterion as the Intersection-over-Union (IoU) between $\hat{\mathcal{A}}^r_{actv}$ and $\mathcal{B}$:

$$\mathcal{M}^{IoU} = \frac{|\hat{\mathcal{A}}^r_{actv} \cap \mathcal{B}|}{|B| + |\hat{\mathcal{A}}^r_{actv} \cap \neg\mathcal{B}|} \tag{11}$$

We will further argue that optimizing for IoU is equivalent to optimizing for F1 in the context of our *PUSH* algorithm. Using the same notation, we write down F1 between $\hat{\mathcal{A}}^r_{actv}$ and $\mathcal{B}$ as follows:

$$\begin{aligned}
\mathcal{M}^{F1} &= \frac{2|\hat{\mathcal{A}}^r_{actv} \cap \mathcal{B}|}{|B| + |\hat{\mathcal{A}}^r_{actv} \cap \mathcal{B}| + |\hat{\mathcal{A}}^r_{actv} \cap \neg\mathcal{B}|} \\
&\equiv \frac{|\hat{\mathcal{A}}^r_{actv} \cap \mathcal{B}|}{|B| + |\hat{\mathcal{A}}^r_{actv}|}
\end{aligned} \tag{12}$$

In order to prove the equivalence of the two criteria, $\mathcal{M}^{IoU}$ and $\mathcal{M}^{F1}$, we simply need to show that when a push operation causes one criterion to increase, the other criterion must also increase as well.

Assume a single push operation reduces $|\hat{\mathcal{A}}^r_{actv} \cap \mathcal{B}|$ from $a_1$ to $a_2$ and reduces $|B|$ from $b_1$ to $b_2$. Let $C = |\hat{\mathcal{A}}^r_{actv}|$, which holds constant throughout the search. Eq.13 shows that $\mathcal{M}^{F1}$ increases iff $\mathcal{M}^{IoU}$ increases.

---

Empirically, we find that the order of $[L, B, R, T]$ does not affect performance. We ran our bbox inference procedure on the Refcocog_val(umd) 10 times, randomly permuting the order of $[L, B, R, T]$ at each *PUSH* iteration. For 90% samples the algorithm returned the same predicted bbox all 10 times. For the remaining samples, the 10 returned bboxes have a joint IoU=0.95±0.07.

We ignore the scaling by 2 in the numerator of $\mathcal{M}^{F1}$ because a constant scalar will not affect the search procedure.

$$
\begin{aligned}
& \frac{a_1}{b_1 + C} < \frac{a_2}{b_2 + C} \\
\leftrightarrow \quad & a_1 b_2 + a_1 C < a_2 b_1 + a_2 C \\
\leftrightarrow \quad & a_1 b_2 + a_1 C - a_1 a_2 < a_2 b_1 + a_2 C - a_1 a_2 \\
\leftrightarrow \quad & \frac{a_1}{b_1 + C - a_1} < \frac{a_2}{b_2 + C - a_2}
\end{aligned}
\tag{13}
$$

## E Different Initialization Methods & Unimodal Pretraining

The figure A3 compares **VLC**$_{\text{Base}}$ with different initializations. We show that most of our performance gain comes from VL pretraining (VLP), while MAE initialization improves the training speed. **(1) Supervised INT1K.** It has a strong prior (67.19% before VLP), but a flat learning curve (+3.36% gain). We hypothesize that such initialization is more biased to predefined INT1K classes, and inconsistent with the visual embedding space learned by MIM. This makes the model difficult to learn multimodal representations by simply switching one objective to another. **(2) Random.** It has the most substantial improvement (+12.23% gain), but still lags behind MAE initialization. We hypothesize that 200K steps (80 epochs) are not sufficient for images. As a comparison, MAE is trained with 1600 epochs on INT1K. More training may mitigate the gap. (3) Neither models, random nor MAE initialization are saturated.

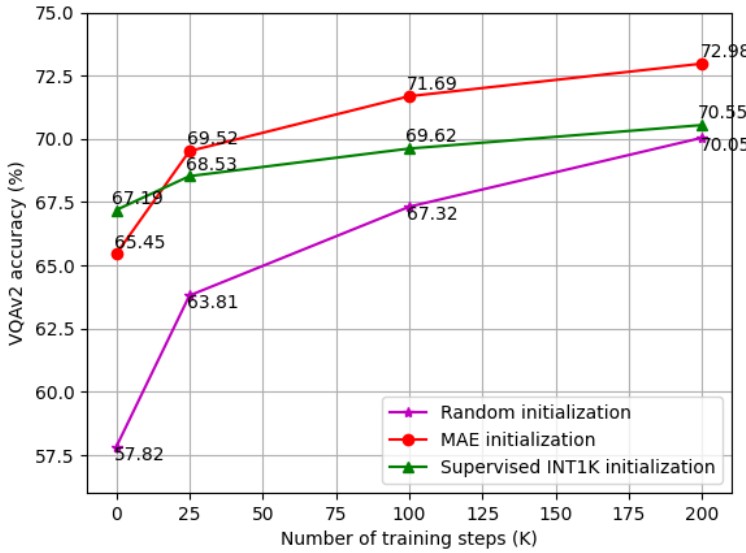

Figure A3: Comparisons of our **VLC**$_{\text{Base}}$ with different initializations. We show that MAE initialization improves the training speed.

Unimodal pretraining initializes the representation space, exposing or collapsing unique properties/clusters. We will expand this analysis, but also include three additional benefits. The table A5 compares unimodal pretraining approaches on VQA. (1) There is a general trend that better pretraining provides stronger downstream priors (METER with Swin/INT21K: 72.38% before VLP), (2) Pretraining can improve the multimodal training speed. As the image/text modalities have different learning speeds. MAE and BERT require 1600 vs 40 epochs on INT1K and a text corpus, respectively. Better image training creates more meaningful representations which allows for "fast mapping" in language acquisition (Fig. 5), and (3) Primary performance gains (+10.2%) come from VLP which demonstrates the effectiveness of our model.

| Models | **VLC**$_{\text{Large}}$ (Ours) | ViLT (Kim et al., 2021) | METER (Dou et al., 2022a) |
|---|---|---|---|
| Initialization | MAE | ViT(21K) | Swin(21K)/Roberta |
| w/o VLP | 66.75 | 67.19 | 72.38 |
| w/ VLP | 76.95 (+10.2) | 71.26 (+4.07) | 76.43 (+4.05) |

Table A5: Comparisons of the benefit of different unimodal pretraining on VQA. We show that most of our performance gain comes from VL pretraining.

## F  Additional Visualizations

We provide additional visualizations for patch-token alignment resulted from pre-training. Specifically, we choose tokens to be nouns in Figure A4, adjectives in Figure A5, and verbs in Figure A6.

The inductive bias of organizing patch-token affinity patterns into bounding boxes can be correctly learned through finetuning, seamlessly adapting the pretrained model to localization tasks. However, in downstream applications when only a single object is expected to be aligned to a phrase (i.e. a span of tokens), model predictions are occasionally distracted by attributes or relations. Table A7 uses the same input image to illustrate the model's sensitivity to varying attributes and relations (after finetuning). The original referring expression was "man walking with purple shirt woman", to which the model gave an incorrect answer. We identify two potential sources of error. First, the model might not be able to detect the woman or the man because they are in the background. Second, the model was able to detect background entities, but the reasoning step might fail to distinguish between entities. We linguistically vary the input expression in order to single out the error source.

Table A7, Middle, indicates that detection error was not the cause of an incorrect prediction, because both the woman and the man can be successfully detected via more accurate and distinct attributes. Table A7, Right suggests that the reasoning step could go wrong for subtle reasons. For example, viewing the scene as a 2D image, the blue shirt man is indeed walking *next to* the purple shirt woman. But it requires nuanced reasoning about the depth or the head direction to infer that the blue shirt man could not be *with* the purple shirt woman. Moreover, both "black shirt" and "purple shirt" occurred twice in the image. Therefore, they could possibly confuse the model. On the other hand, "white cap" and "yellow shorts" are distinct attributes, which easily help the model with making the correct choice.

## G  Additional Results and Error Analysis for Downstream Grounding Tasks

Table A6 compares the Referring Expression Comprehension performance on both the validation and test splits. Results on the test splits exhibit a similar trend as validation splits. Our performance is superior to the previous specialized models, on par with moderate-size unified models, but is lower than those very largest and slowest unified models.

Next, we analyze common prediction errors which could inform future research on the unsolved difficulties of Referring Expression tasks. Table A8 visualizes the error-prone cases on Referring Expression Comprehension. The model blunders when the target entity is specified through complex attributes or relations. The model also occasionally confuses a head noun with its modifiers, or gets distracted by an irrelevant entity sharing a mentioned attribute. Mistakes can also stem from inadequate text-reading ability from visual stimuli.

Table A9 shows a performance breakdown according to four potential sources of difficulty. Table A9a suggests that the model particularly struggles when a referring expression contains multiple noun chunks. We believe that the number of noun chunks is a reasonable indicator of difficulty because it also correlates well with the sentence length and linguistic complexity. Table A9b suggests that the model is less performant when it encounters an unfamiliar head noun. We parse the expressions and extract noun chunks via SpaCy. A token is identified as the head noun if its part-of-speech tag is one of ['NOUN', 'PROPN'] and its role in the dependency tree is one of ['nsubj', 'root']. The head noun frequencies are counted across Refcoco/+/g_val.

---

These heuristics occasionally fail to identify a head noun or identify more than one head noun in $< 5\%$ of the cases.

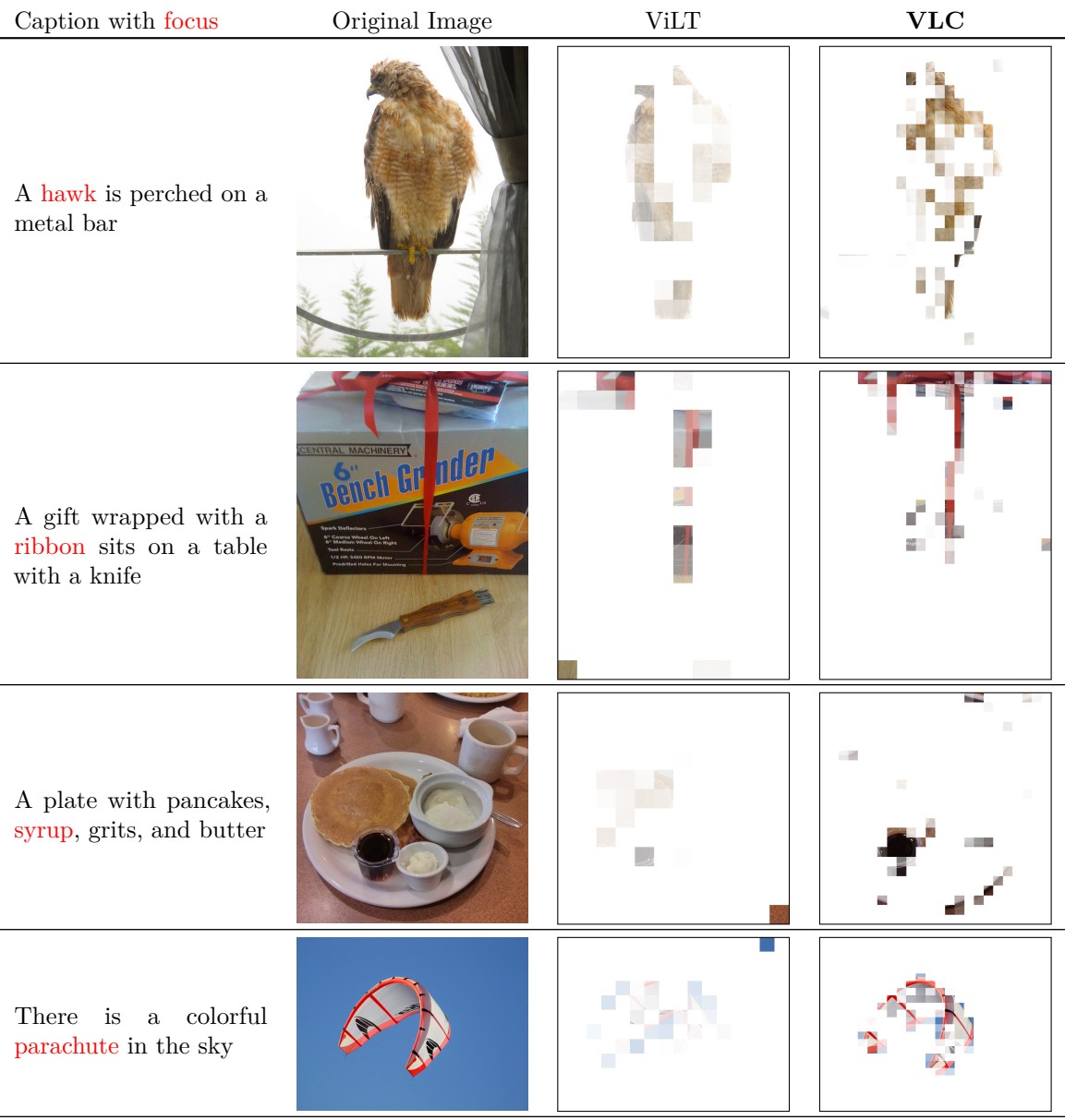

Figure A4: We visualize patch similarities with nouns on OOD images. Note that ViLT is often picking up on relevant features but has the strongest correlation with a single, presumably predictive patch.

Table A9c plots the performance against the area of the ground-truth bounding box. We observe that the performance drops drastically on extremely small regions. Table A9d plots the (zero-out) segmentation performance against the convexity of the ground-truth polygon mask. We measure convexity as the ratio between the area of the polygon and the area of its convex hull. Note, the presence of holes inside a mask or non-contiguity could also lead to a small convexity score. Thus, our findings reflect ample room for improvement on localizing small and less convex regions.

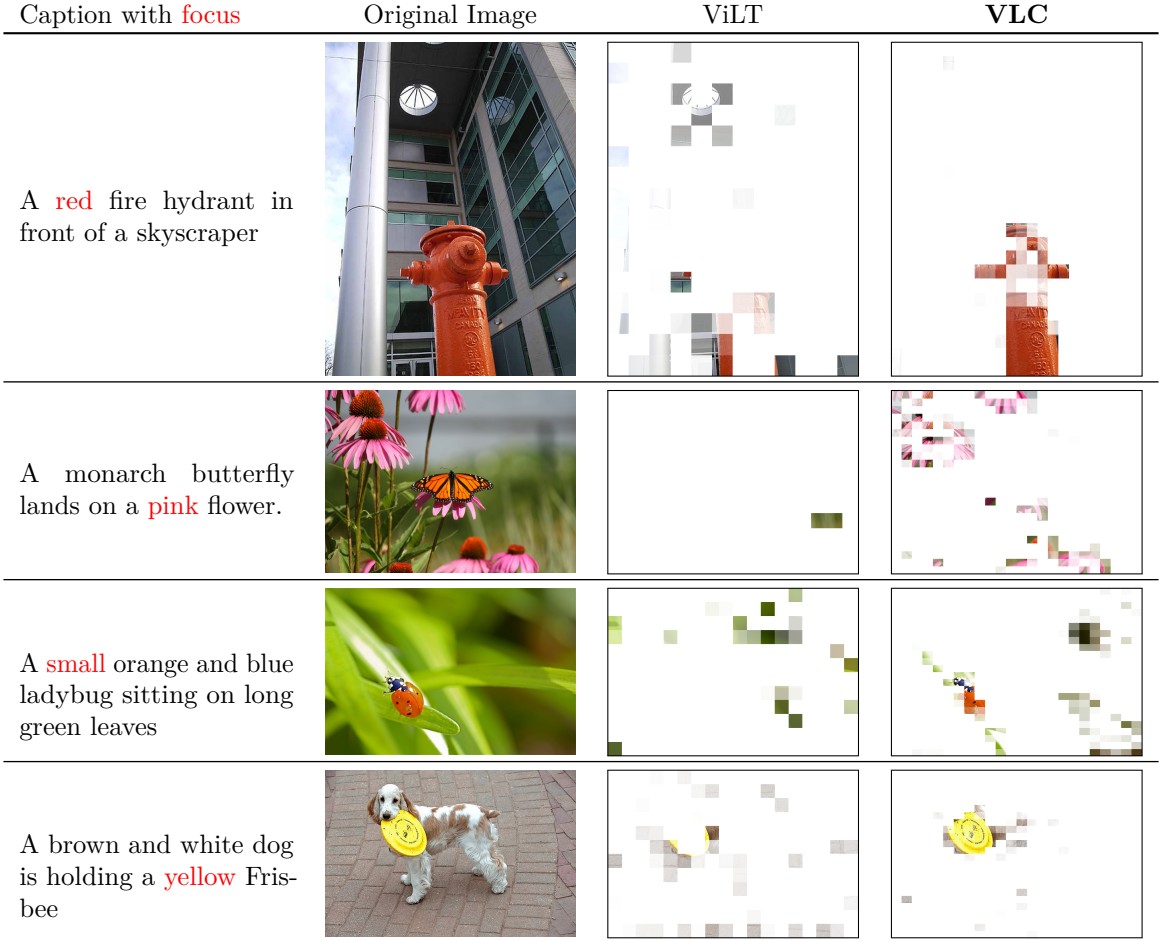

Figure A5: We visualize patch similarities with adjectives on OOD images. **VLC** produces more accurate and comprehensive alignment. Note that the lady bug is correctly associated with the relative size *small*. Future work would ideally investigate how models align visual entities to more abstract and comparative concepts in the language domain.

## H   Limitations and Societal Impact

In our work, we train our model with image-text pairs of 4M/5.6M unqiue images. Limited to our computation resource, it is unclear how the model performance will be with further scaling. In addition, we only focus on images with English descriptions. The generalization ability to other languages is left for the futher work.

While our paper shows promising results on vision-language pretraining and the scaling of training data and model size, additional analysis should be undertaken to assess the quality of pretraining web data before further deployment. Because web data may contain gender/racial bias, hate speech, private information and other unsuitable images. Only optimizing the performance may have unwanted model behaviors.

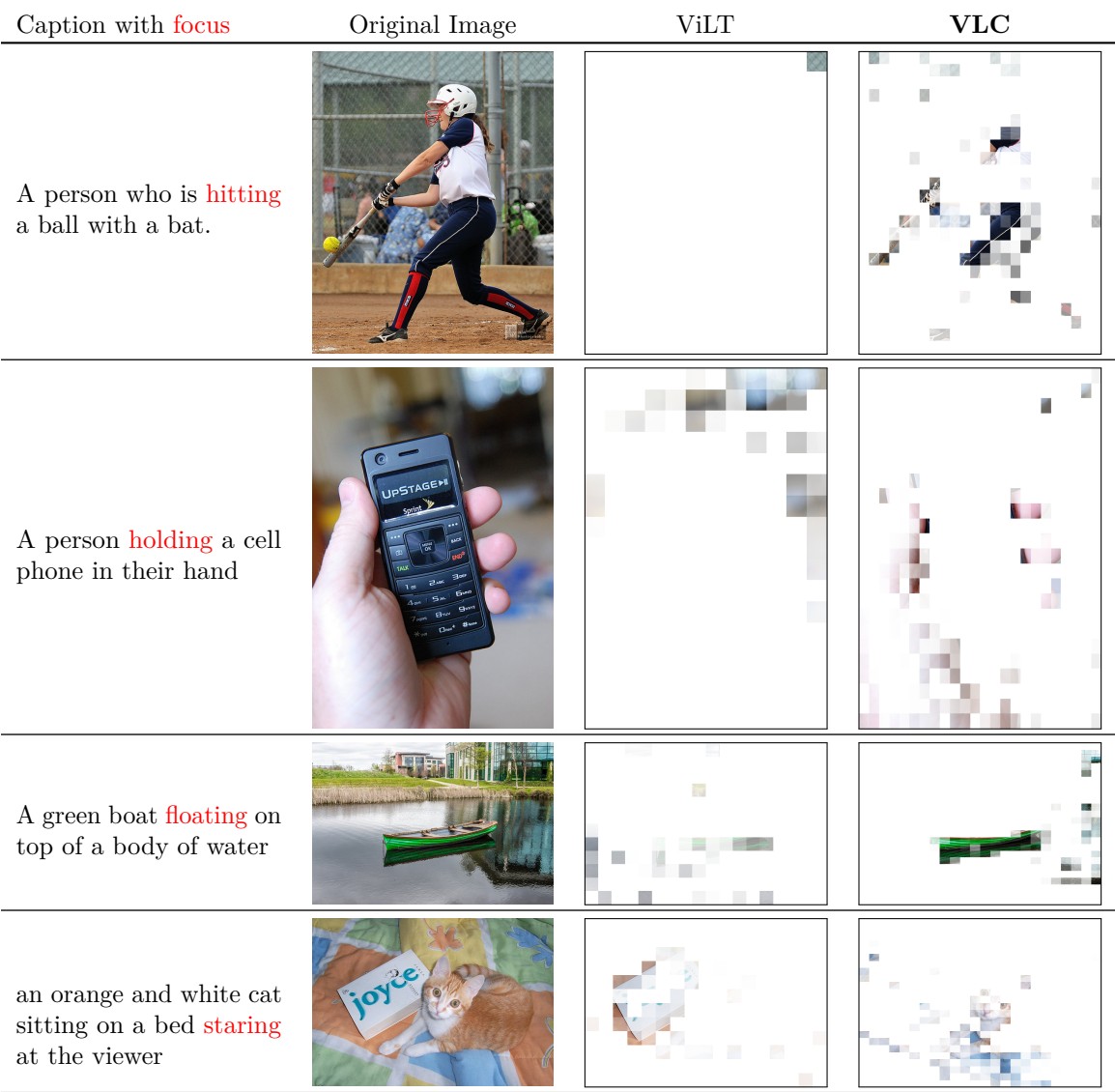

Figure A6: We visualize patch similarities with verbs on OOD images. Note that aligning verbs to static images is a slightly strange task, but our model still manages to associate key perceptual indicators to a verb's semantics. For example, *holding* is aligned to a person's hand and *staring* picks up on the cat's eyes.

| | Model | Time | Params | Refcoco | | | Refcoco+ | | | Refcocog(umd) | |
|---|---|---|---|---|---|---|---|---|---|---|---|
| | | | | val | testA | testB | val | testA | testB | val | test |
| *Modular* | VL-T5 (Cho et al., 2021) | 5.9x | 290M | — | — | — | — | — | — | 71.2 | 71.3 |
| | TransVG (Deng et al., 2021) | 2.8x | 169M | 80.83 | 83.38 | 76.94 | 68.00 | 72.46 | 59.24 | 68.71 | 67.98 |
| | TransVG++ (Deng et al., 2022) | — | — | 86.28 | 88.37 | 80.97 | 75.39 | 80.45 | 66.28 | 76.18 | 76.30 |
| | QRNet (Ye et al., 2022) | 5.5x | 273M | 84.01 | 85.85 | 82.34 | 72.94 | 76.17 | 63.81 | 73.03 | 72.52 |
| | RefTrans (Li & Sigal, 2021) | — | — | 85.65 | 88.73 | 81.16 | 77.55 | 82.26 | 68.99 | 79.25 | 80.01 |
| | SeqTR (Zhu et al., 2022) | 4.4x | 108M | 83.72 | 86.51 | 81.24 | 71.45 | 76.26 | 64.88 | 74.86 | 74.21 |
| | UNITER$_{\text{Large}}$ (Chen et al., 2020b) | — | 371M | 81.41 | 87.04 | 74.17 | 75.90 | 81.45 | 66.70 | 74.86 | 75.77 |
| | VILLA$_{\text{Large}}$ (Gan et al., 2020) | — | 371M | 82.39 | 87.48 | 74.84 | 76.17 | 81.54 | 66.84 | 76.18 | 76.71 |
| | **VLC**$_{\text{Base}}$ (ours) | 1.0x | 110M | 86.67 | 89.15 | 81.99 | 77.44 | 82.57 | 69.71 | 80.43 | 80.22 |
| *Unified* | OFA$_{\text{Medium}}$ (Wang et al., 2022a) | — | 93M | 85.34 | 87.68 | 77.92 | 76.09 | 83.04 | 66.25 | 78.76 | 78.58 |
| | MDETR-R101 (Kamath et al., 2021) | 3.0x | 185M | 86.75 | 89.58 | 81.41 | 79.52 | 84.09 | 70.62 | 81.64 | 80.89 |
| | MDETR-ENB3 (Kamath et al., 2021) | 2.6x | 153M | 87.51 | 90.40 | 82.67 | 81.13 | 85.52 | 72.96 | 83.35 | 83.31 |
| | UniTAB (Yang et al., 2022) | 19.4x | 198M | 88.59 | 91.06 | 83.75 | 80.97 | 85.36 | 71.55 | 84.58 | 84.70 |
| | OFA$_{\text{Base}}$ (Wang et al., 2022a) | 6.0x | 180M | 88.48 | 90.67 | 83.30 | 81.39 | 87.15 | 74.29 | 82.29 | 82.31 |
| | OFA$_{\text{Large}}$ (Wang et al., 2022a) | 11.5x | 470M | 90.05 | 92.93 | 85.26 | 85.80 | 89.87 | 79.22 | 85.89 | 86.55 |
| | OFA (Wang et al., 2022a) | — | 930M | 92.04 | 94.03 | 88.44 | 87.86 | 91.70 | 80.71 | 88.07 | 88.78 |

Table A6: Results on Referring Expression Comprehension. Models are compared along two axes: efficiency-performance and efficiency-versatility. Our model beats or competes with other models, while being much lighter and faster. The Params column counts *all* parameters required for inference, including the upstream RoI extractor (if needed), the main architecture, and specialized output heads.

| Input image | Predicted (incorrect) | Predicted (correct) |
|---|---|---|

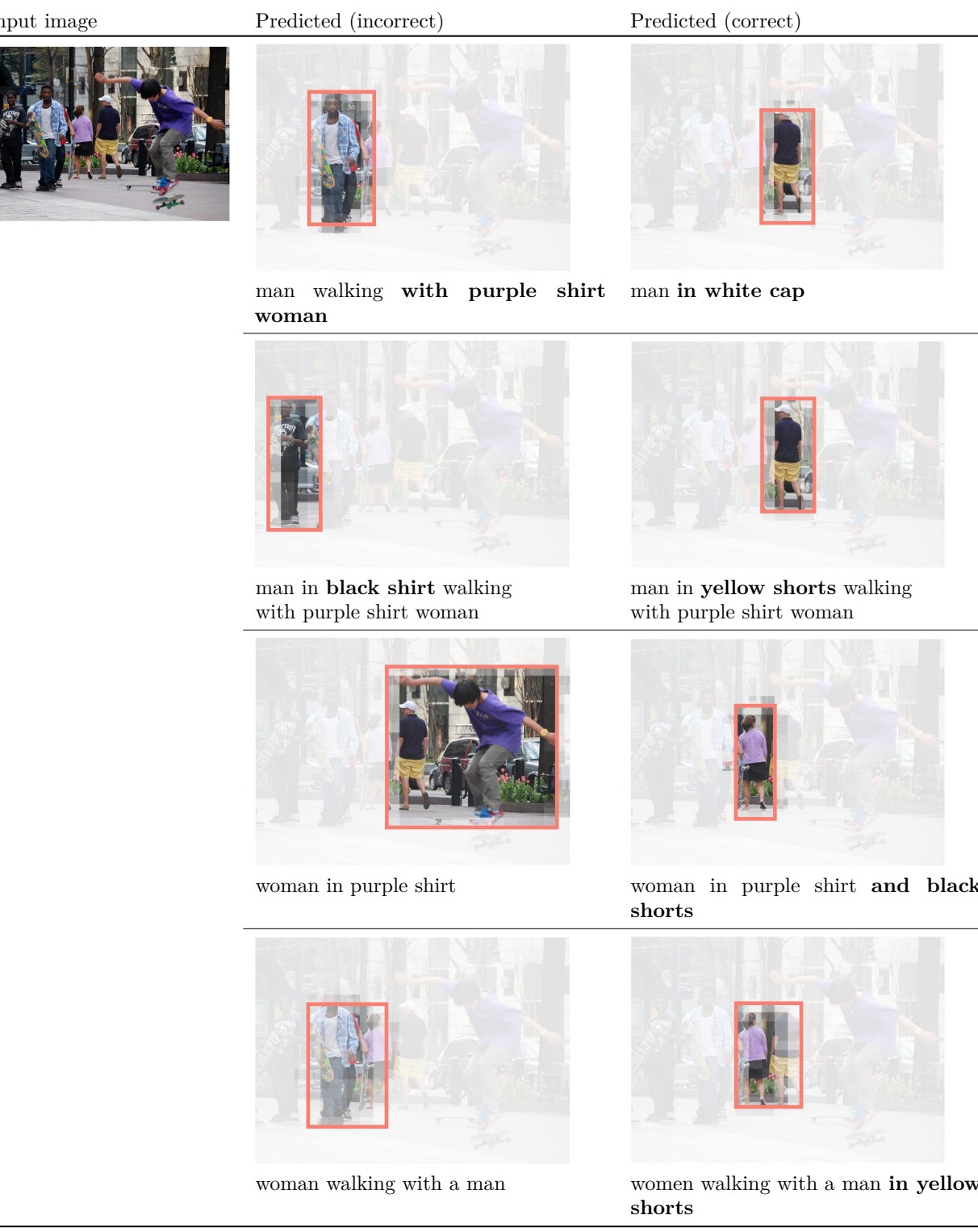

Table A7: Qualitative demonstration: The model is able to detect small entities in the background. But the model's success at distinguishing between entities relies on accurate and distinct attributes or relations. Here we juxtapose changes to phrasing or addition of a clause.

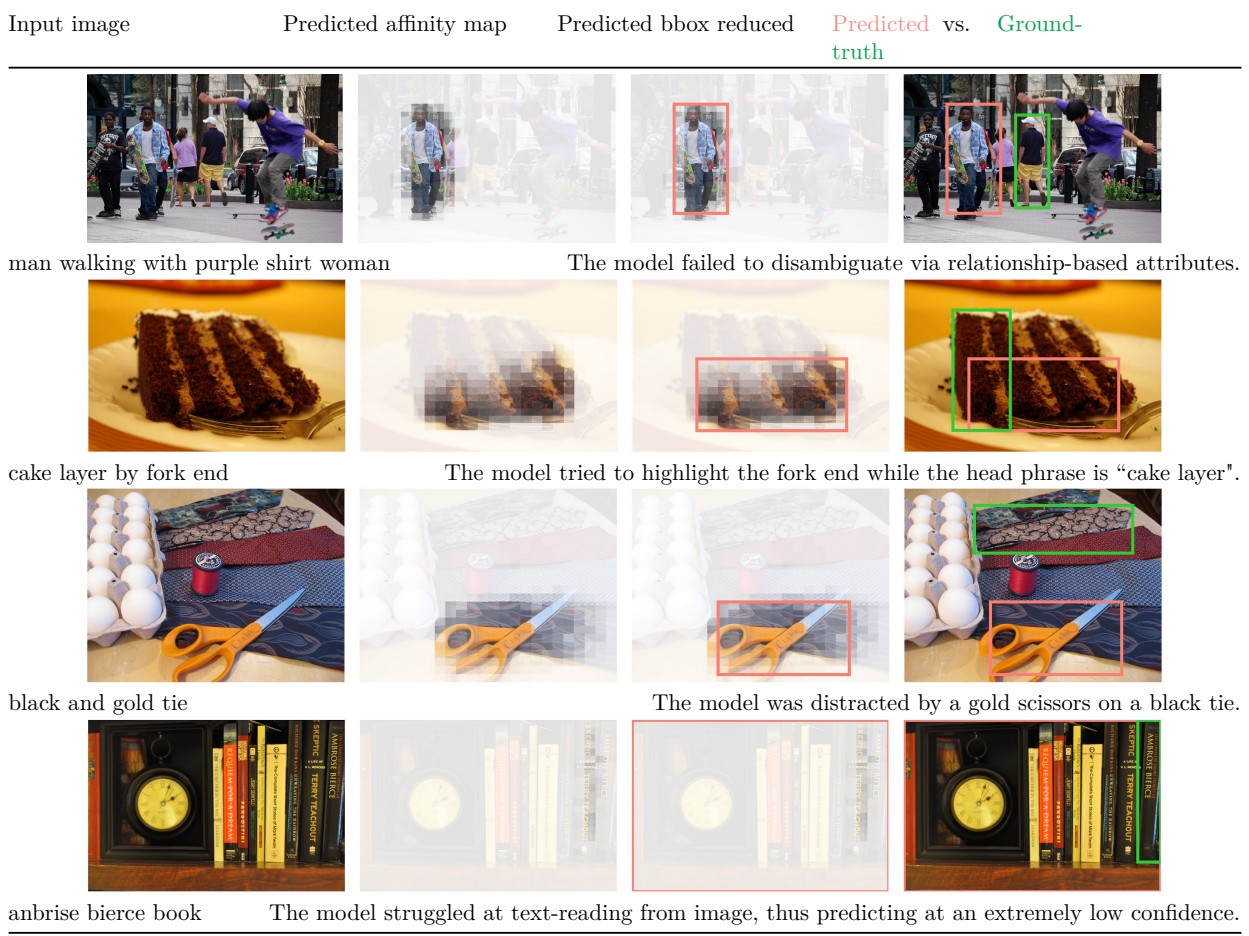

Table A8: Qualitative demonstration. Common error types include 1) failure to resolve relationship-based attributes, 2) incorrect head-noun identification, 3) distraction by an irrelevant object sharing a mentioned attribute, and 4) inadequate text-reading from visual stimuli.

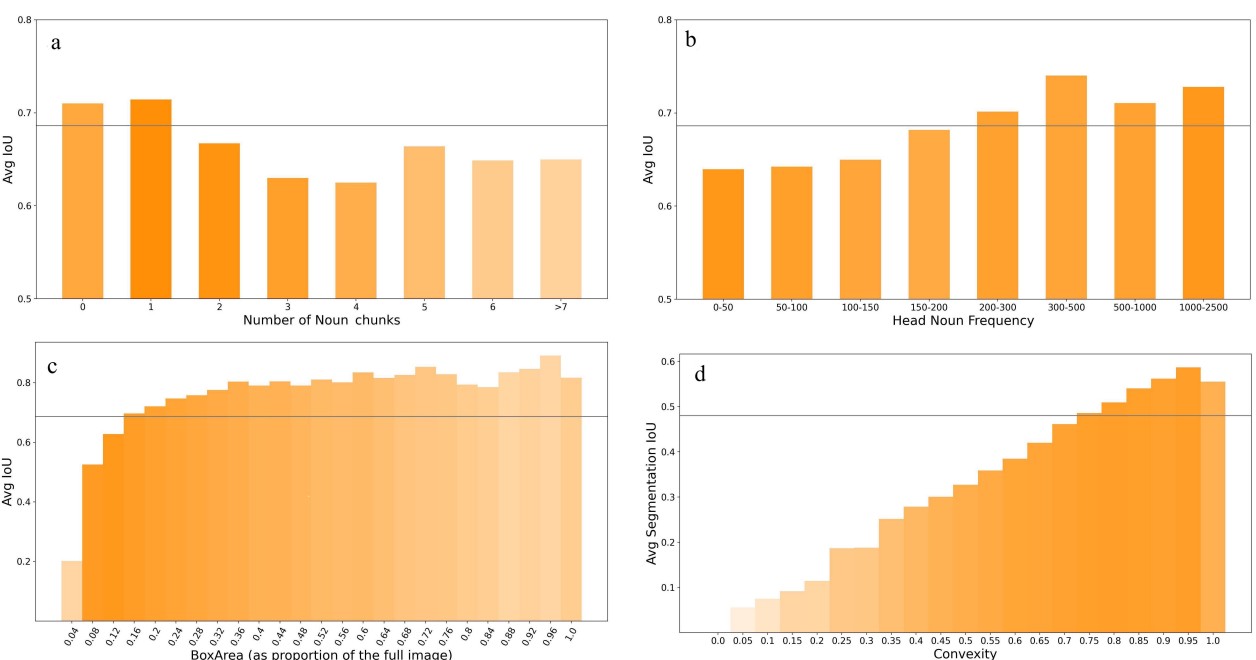

Table A9: Performance breakdown according to different indicators of the difficulty level. We observe that the performance drops when a) the referring expression contains multiple noun chunks, b) the referring expression has a rare head noun, c) the ground-truth bounding box has a small area, d) the ground-truth polygon mask is less convex. Statistics are computed across Refcoco/+/g_val. Shades indicate the number of examples in each bin. The horizontal line marks the global performance averaged over all bins.

