# OpenReview forum: "Training Vision-Language Transformers from Captions"
_TMLR — Accepted by TMLR_

### Review · Reviewer_2YtN · 2023-07-25

**Summary Of Contributions:**

This paper proposes a model for vision + language learning using only captioning data. The approach is evaluated on a set of standard tasks and datasets.

**Audience:**

Yes

**Claims And Evidence:**

No

**Requested Changes:**

I think the claims should be revised and clarified with respect to the prior works. The current claims aren't entirely true as many recent works use similar models, tasks, and data for vision + language learning without using ROIs or object level data.

Many related works are omitted and should be discussed (coca, flamingo, simvlm, etc).

**Strengths And Weaknesses:**


I feel like the abstract is incorrect. "Existing work, whether explicitly utilizing bounding boxes
(Chen et al., 2020b; Tan & Bansal, 2019; Lu et al., 2019) or patches (Kim et al., 2021),
assumes that the visual backbone must first be trained on ImageNet (Russakovsky et al.,
2015) class prediction before being integrated into a multimodal linguistic pipeline"

There are many works that use weak labels or captions for training a ViT backbone. For example PaLI (Chen et al ICLR 2022) uses caption data to train the vision model. CoCA (Yu et al 2022) and SimVLM (Wang et al 2021) also use captioning data to train the ViT. These works are not addressed in the paper. Works like CLIP (Radford et al 2021) and Flamingo (Alayrac et al 2022) also do not use imagenet pre-training.

The claimed contributions:
" 1. Smaller and faster at inference, 2. Avoiding use of ROIs, and 3. Not leveraging
object-level supervised labels for pretraining."

Aren't that strong. Using ROIs, is no longer standard, pretty much all recent works have moved away from using ROIs and object-level supervision.

The approach itself (e.g., Figure 2), is nothing novel. It consists of modality encoders, concatenation, then a transformer followed by multiple output task decoders. There are many works using this type of model. This work, with its pretraining tasks, seems very similar to many previous works (e.g., arXiv:2205.00949).

Methods like CoCa are missing from Table 2. With so many missing related works and missing comparisons, it is hard to tell if the state of the art comparisons are accurate. Table 3 results are quite weak as well, especially since the main advantage of using only weak labels is the ability to scale to larger datasets, the comparisons with other similar methods (e.g., Pali, coca, flamingo, etc) should be better.

The localization results are more interesting, showing the models ability to adapt to a localization task. I think focusing on this aspect of the paper would be better, as it is more unique and different from many previous works.

---

### Review · Reviewer_SRjs · 2023-07-26

**Summary Of Contributions:**

This paper presents a method to learn vision-language Transformers with image-text pairs and a mask modeling pretext task. Without using classification or detection labels, the model achieves good performance on various vision tasks. The paper also shows a clear performance improvement over ViLT.

**Audience:**

Yes

**Claims And Evidence:**

Yes

**Requested Changes:**

Please refer to my comments above. I think it would be beneficial to include and discuss some more recent methods.

**Strengths And Weaknesses:**

Strengths:

- Extensive experiments are conducted to verify the effectiveness of the proposed method. The proposed method can lead to good grounding performance and benefit compositional reasoning.

- The proposed method is simple yet effective. The model is also more efficient compared to existing ones.

Weaknesses:

- The claim in the Introduction "Our Vision-Language from Captions (VLC) model matches or outperforms nearly all vision-language transformers" is not well-supported. There are quite a few recent works on learning vision-language Transformers including [r1, r2, r3]. Since the existing methods considered in the paper are a bit out-of-date, I think it would be better to modify some claims accordingly.

[r1] UL2: Unifying Language Learning Paradigms

[r2] PALI: A JOINTLY-SCALED MULTILINGUAL LANGUAGE-IMAGE MODEL

[r3] CLIPPO: Image-and-Language Understanding from Pixels Only

- Some conclusions are a bit confusing. The paper claims the proposed method benefits from avoiding a vision-only processing stack and avoiding priors from ImageNet pretraining. But the results in the paper show the proposed model can achieve comparable performance without these designs or pretraining. Is it possible that the model can be stronger if a stronger (and large) vision tower is added to the proposed framework? Considering there are many exciting progresses like CLIP on learning better backbone models, standing on the shoulders of these powerful models instead of intentionally avoiding using them may be a promising solution to solve vision problems.

---

### Review · Reviewer_PotT · 2023-07-29

**Summary Of Contributions:**

The paper proposes a (pre)training method for vision language transformer that does not require a pretrained image-encoder trained either on Imagenet or "dense" labels such as bounding box annotation, region annotation etc. The new method is based on masked __ modeling for both images and text which is shown to be effective in learning good representation without any pre-pretraining required. Other interesting properties also emerge, such as the fact that the "traditional" model seems to provide a good headstart but also a lower "ceiling" compared to the proposed model when the network sizes grow. The results show competitive or better results compared to "traditionally" trained multimodal transformers when compared against a wide range of existing methods.

**Audience:**

Yes

**Claims And Evidence:**

Yes

**Requested Changes:**

[Q1]: Ref W2, Please clarify the scope and importance of being able to do a specific set of conditions (train without Imagenet pretraining on a small amount of data). Please also clarify what conceptual differences are there compared to CLIP-like (Hyper or CoCa-like models.

[Q2]: Why is >10M data treated as a separate thing? Is the claim that the proposed method is more scalable? Is the paper claiming faster convergence on smaller data? Is it better than initializing from a previously trained model such as CLIP? Please clarify what "value proposition" is offered by the paper.


**Strengths And Weaknesses:**

Strengths:

[S1]: Clearly written: The paper is clearly written and contains sufficient details and thorough descriptions of the experimental design. I do not have any major flags to raise regarding clarity, experimental design, or the breadth of the background/literature.

[S2]: Technically sound and of interest to the community: The paper clearly satisfies both big-ticket criteria for TMLR, i.e., Technical soundness and interest to the community. To this reviewer, the simple-yet-effective model design, thorough experiments, and great writing all clearly show the technical soundness of the paper. Similarly, while there are caveats (see below), the core set of findings in indeed novel and should be of interest to (at least some) member of the community.


Weaknesses:

[W1]: Core premise unclear: The core premise of the paper appears to be that it is possible to train transformers from text alone unlike prior methods which requires training with region/box annotations and/or imagenet trained initialization. As a researcher also working in this field, I have not interpreted the state of multimodal pretraining in this light. To me, the choice to use box annotation and/or Imagenet pretraining was to speed up training and/or minimize training stability (i.e., done for practical reasons) and not because the community collectively thought that training from scratch was not possible. However, I do admit, after this manuscript that there does seem to be a dearth of "training from scratch" models. So, the findings are still valuable. Furthermore, the paper does show that it is possible to get competitive results without the use of additional data, which is indeed valuable. However, I would like to see more discussion/proof about the core premise and exactly what the authors are claiming so that the reviewers can evaluate it in context.

[W2]: Rather niche/cherrypicked set of conditions: To this reviewer, the final set of conditions seems to be ever-tightening to the point that it becomes quite rigid/artificial.
- It is possible to train transformers with text-only supervision
(So does Virtex, https://arxiv.org/pdf/2006.06666.pdf), which has a transformer for the text side but not of vision side which still uses CNNs.

- So, revising that, I guess the paper is claiming that *both* vision and text transformers can be trained with text only. But, CoCa and CLIP also show that too. Then, the paper makes another clarification that they use >10M data. To this point, CLIP (and variants) and CoCa both do continue to scale with more data, but both have been shown to *start* working with a much smaller amount of data (~15m in DeCLIP, SLIP, etc..12M in MERU ).
- So, I think the paper is showing that vision and text transformers can be trained without Imagenet initialization on a small data regime.

Is this understanding correct?

To me, this is neither made adequately clear in the paper nor do I see a broad appeal for this set of conditions. Why are they important? Together with W1, I would say that this lack of motivation/clarity about the scope (also W1) of the paper is its biggest weakness.

---

### Decision · Action_Editors · 2023-09-01

**Recommendation:** Accept with minor revision

**Comment:**

The submission presents a method building on masked autoencoders called Vision-Language from Captions (VLC) for training vision-language models using only caption data.

Given the modifications promised by the authors, the paper would meet the bar for acceptance in terms of empirical support of the claims made and appeal to TMLR's community. Acceptance is therefore conditional on the following modifications appearing in the camera-ready version:

* (2YtN) Edit the abstract to make our contributions clear
* (2YtN) Clarify our motivation of controlling for the dataset size and initialization
* (PotT, SRjs) Add comparisons with CoCA and SimVLM on model and training data scales
* (SRjs) Add conceptual comparisons with PaLI, CLIPPO and Flamingo in related works
* (PotT) Add conceptual comparisons with contrastive models such CLIP, CoCA in related works
* (PotT,2YtN,SRjs) Add comparisons with recent models on GLUE evaluations
* Add missing references such as VirTex, UL2, CLIPPO
* Address Reviewer 2YtN's concern that "the linear probing [in the added results on ImageNet] show [CoCa/CLIP/etc.] to be better" and determine whether this is a case of overfitting or better hyperparameter tuning in the finetuning case.

**Audience:**

The findings presented in the submission is of interest to at least some individuals in TMLR's audience. Reviewer PotT notes the paper appears to be "showing that vision and text transformers can be trained without Imagenet initialization on a small data regime". This narrows the appeal to TMLR's audience (a concern shared by Reviewer 2YtN and which manifests itself for instance in the scope of competing approaches considered), but Reviewer PotT still finds that "the core set of findings in indeed novel and should be of interest to (at least some) member of the community" and Reviewer 2YtN finds that the submission "[demonstrates that pretraining a vision language model without any supervised data can be done competitively with models that use some supervised pretraining data" and "[presents] a single multimodal encoder architecture and shows it works for various tasks".

**Claims And Evidence:**

The proposed approach (VLT) is compared against ViLT and other competing approaches on several tasks (zero-shot visual reasoning on Kilogram, text/image retrieval on Flickr30K (1K)/MSCOCO (5K), image-text understanding on VQAv2/NLVR, and image-text grounding). Ablations are presented on several aspects of VLC, and the submission investigates its behavior is influenced by its design choices.

Reviewer SRjs points out that "[there] are quite a few recent works on learning vision-language Transformers" which are not considered in the paper, which makes the claim that "our Vision-Language from Captions (VLC) model matches or outperforms nearly all vision-language transformers" not well-supported. The authors respond that they "only consider the vision-language transformers with comparable model sizes and training data" and promise to clarify and justify this choice in the updated manuscript. This addresses the issue by virtue of appropriately scoping the claims.